# Mitochondrial translation failure represses cholesterol gene expression via Pyk2-Gsk3$\beta$-Srebp2 axis

Takahiro Toshima[1,2], Mikako Yagi[1,2] , Yura Do[1], Haruka Hirai[1,2], Yuya Kunisaki[1], Dongchon Kang[1,3,4], Takeshi Uchiumi[1,2]

**Neurodegenerative diseases and other age-related disorders are closely associated with mitochondrial dysfunction. We previously showed that mice with neuron-specific deficiency of mitochondrial translation exhibit leukoencephalopathy because of demyelination. Reduced cholesterol metabolism has been associated with demyelinating diseases of the brain such as Alzheimer's disease. However, the molecular mechanisms involved and relevance to the pathogenesis remained unknown. In this study, we show that inhibition of mitochondrial translation significantly reduced expression of the cholesterol synthase genes and degraded their sterol-regulated transcription factor, sterol regulatory element-binding protein 2 (Srebp2). Furthermore, the phosphorylation of Pyk2 and Gsk3$\beta$ was increased in the white matter of p32cKO mice. We observed that Pyk2 inhibitors reduced the phosphorylation of Gsk3$\beta$ and that GSK3$\beta$ inhibitors suppressed degradation of the transcription factor Srebp2. The Pyk2–Gsk3$\beta$ axis is involved in the ubiquitination of Srebp2 and reduced expression of cholesterol gene. These results suggest that inhibition of mitochondrial translation may be a causative mechanism of neurodegenerative diseases of aging. Improving the mitochondrial translation or effectiveness of Gsk3$\beta$ inhibitors is a potential therapeutic strategy for leukoencephalopathy.**

## Introduction

Mitochondria are eukaryotic intracellular organelles that perform a variety of functions, including not only ATP production through oxidative phosphorylation but also lipid metabolism, signaling to induce apoptosis, and the generation of reactive oxygen species (Wengrod & Gardner, 2015). Mitochondria are important organelles for intracellular energy production, and their dysfunction readily affects the central nervous system and cardiac muscle, which contain many mitochondria (Wallace, 1999). Inhibition of mitochondrial translation is thought to cause functional defects in all mtDNA-encoded proteins, leading to severe respiratory chain dysfunction. Thus, alterations in the expression of mtDNA-encoded proteins are thought to be associated with diseases such as mitochondrial diseases (Boczonadi & Horvath, 2014).

Mitochondrial-to-nuclear retrograde signaling plays a crucial role in regulating gene expression in the nucleus in response to mitochondrial dysfunction. Furthermore, the decline of mitochondrial function over time has been shown to be associated with age-related diseases. p32, which is also known as Complement C1q Binding Protein, is mainly localized in the mitochondrial matrix. Previously, we found that p32-deficient mice exhibit embryonic lethality and that p32-deficient MEFs exhibit severe mitochondrial respiratory chain dysfunction because of severely impaired mitochondrial protein synthesis. Furthermore, p32 binding to mitochondrial RNA and mitochondrial ribosomes correlates with mitochondrial translation, suggesting that p32 is an RNA and protein chaperone required for functional mitochondrial ribosome assembly (Yagi et al, 2012). Conditional p32KO mice with neuron-specific impairment of mitochondrial translation function have been reported to exhibit brain demyelination, leukoencephalopathy development, and mortality within just 8 wk (Yagi et al, 2017). Myelination in the central nervous system is typically carried out by oligodendrocytes; in mice, this process commences 10 d after birth (Kuboyama et al, 2016). We also discovered decreases in sphingomyelin and phosphatidylcholine—both components of the myelin sheath—in the brains of mice with neuron-specific conditional p32KO (Yagi et al, 2017). Additionally, several studies have suggested a link between mitochondrial abnormalities and neurodegenerative conditions, such as Alzheimer's disease and Parkinson's disease (Wallace et al, 1988; Kuramoto et al, 2011; Meng et al, 2017; Gulen et al, 2023; Van Acker et al, 2023). Cardiomyopathy was mainly observed in patients with p32 mutations, but other symptoms such as weakness, ptosis, PEO, post-traumatic depression and peripheral nervous system abnormalities were also observed, suggesting that p32 is involved in neurological dysfunction (Jiang et al, 1999). It is thus believed that mitochondrial translation defects cause reduced expression of myelin components and leukoencephalopathy, but the molecular mechanism behind this remains unclear.

[1]Department of Clinical Chemistry and Laboratory Medicine, Kyushu University, Fukuoka, Japan    [2]Department of Health Sciences, Graduate School of Medical Sciences, Kyushu University, Fukuoka, Japan    [3]Kashiigaoka Rehabilitation Hospital, Fukuoka, Japan    [4]Department of Medical Laboratory Science, Faculty of Health Sciences, Junshin Gakuen University, Fukuoka, Japan

Correspondence: uchiumi.takeshi.008@m.kyushu-u.ac.jp

Most of the cholesterol in the brain is instead produced by oligodendrocytes and stored in myelin sheaths to maintain axon insulation (Dietschy & Turley, 2004; Chrast et al, 2011). Synaptic vesicles are synthesized by synaptophysin, a protein that is abundant in synaptic vesicle membranes, via endocytosis of the plasma membrane. These findings indicate that cholesterol plays an important role in the development and maintenance of brain function, particularly in nerve conduction.

Sterol regulatory element-binding protein 2 (Srebp2), a sterol regulatory transcription factor, has a crucial role in cholesterol biosynthesis. Srebp2 is synthesized as a precursor and processed into a mature nuclear form (nSrebp2) in the Golgi apparatus before becoming a functional transcription factor. In the nucleus, nSrebp2 regulates the expression of many genes involved in cholesterol homeostasis (Guo et al, 2024). Srebp2 activity has also been reported to be reduced in neurodegenerative diseases such as Alzheimer's disease (Mohamed et al, 2018; Tang et al, 2023), and Srebp2 has been reported to play an important role in the supply of cholesterol required for myelination (Zhou et al, 2021).

Glycogen synthase kinase 3 (Gsk3) is a serine/threonine protein kinase originally identified as a regulator of glycogen metabolism. It is also reported to be an important regulator of neuronal survival (Castaño et al, 2010). Gsk3$\beta$ has also been identified as a potential risk factor for Alzheimer's disease (Beurel et al, 2015). The activity of Gsk3$\beta$ is regulated by Ser9 phosphorylation, which leads to its inactivation, and Tyr216 phosphorylation, which activates Gsk3$\beta$. Previous studies have shown that Gsk3$\beta$ is activated during periods of starvation or insulin deficiency and phosphorylates Ser434 of its precursor SREBP-1c on the ER membrane, leading to its degradation by proteasomes (Bengoechea-Alonso & Ericsson, 2009; Dong et al, 2015).

Proline-rich tyrosine kinase 2 (Pyk2), a member of the Focal Adhesion Kinase family of non-receptor tyrosine kinases, is activated by a $Ca^{2+}$-dependent mechanism (Lev et al, 1995). Calcium regulation is essential for synaptic plasticity and its dysregulation is a hallmark of neurodegenerative diseases. Pyk2 has also been implicated in neurodegenerative diseases such as Alzheimer's and Huntington's disease (Lambert et al, 2013; Giralt et al, 2018; Kilinc et al, 2020). In neurons, Pyk2 has been shown to activate Gsk3$\beta$ (Theos et al, 2005; Dedoni et al, 2024).

Although several transcription factors are involved in the regulation of Srebp2 (Li et al, 2011; Mesquita et al, 2020), how mitochondrial dysfunction affects Srebp2 expression patterns and gene expression related to cholesterol synthesis remains unclear. Our study was implemented to explore the hypothesis that mitochondrial dysfunction may be responsible for demyelination and focused on the sterol regulatory proteins Srebp2, Gsk3$\beta$, and Pyk2 to elucidate the mechanism by which leukoencephalopathy develops in mice with neuron-specific conditional knockout of p32 (p32cKO).

# Results

## p32 deficiency results in demyelination and reduced synaptogenesis

To study the state of synapses in the brain, transmission electron microscopy of the white matter of 5-wk-old mice in the pons and medulla oblongata revealed a decrease in the number of synaptic vesicles present in neurons of p32cKO mice (Fig 1A). Moreover, the area of synaptic vesicles as a percentage of the area of nerve terminals was significantly lower in p32cKO mice (Fig 1B). We also examined by immunohistochemistry using antibodies against synaptophysin, a synaptic vesicle membrane protein in the white matter region of the pons and medulla oblongata of control and p32cKO mice at 6 wk of age. We observed that the synaptophysin staining of p32cKO mice was significantly reduced (Fig S1A). Kluver–Barrera staining showed significant demyelination (destruction of myelin) or hypomyelination throughout the white matter (Fig S1B). Furthermore, a comparison of myelin morphology revealed that a thick layer of myelin sheath was formed around axons in wild-type mice, whereas p32cKO mice had thin myelinated and unmyelinated axons (Fig 1C). These findings suggest that membrane synthesis is restricted in the brains of p32-deficient mice, which ultimately affects the development of synapses and myelin.

## p32 deficiency leads to decreased cholesterol synthase gene expression

We first compared cholesterol synthase gene expression in the white matter region of pons and medulla oblongata between p32cKO and wild-type mice by real-time PCR. In the brain of p32cKO mice, we found that *Srebp2* (*sterol responsive element-binding protein 2*), *Hmgcr* (*3-hydroxy-3-methylglutaryl-CoA reductase*), *Hmgcs1* (*3-hydroxy-3-methylglutaryl-CoA synthase 1*), and seven other cholesterol synthase genes and were significantly downregulated (Fig 1D). These results suggest that the brain with mitochondrial translation defects may exhibit demyelination because of a reduction in the capacity to synthesize cholesterol. As Srebp2 is a master transcription factor for genes in the cholesterol synthesis system (Luo et al, 2020; Xiao et al, 2023), we hypothesized that the reduced expression of this cholesterol gene might be because of reduced expression levels of nuclear active Srebp2. Therefore, we next compared the protein expression levels of Srebp2, a master regulator of cholesterol synthase, in the medulla oblongata, a region of the brain's white matter region of p32cKO, and wild-type mice. We found that the expression level of nSrebp2, which functions as a transcription factor, was significantly decreased in the brains of p32cKO mice compared with that in the brains of wild-type mice (Fig 1E). These findings suggest that, in brains with mitochondrial translation defects, particularly in white matter regions, reduced nSrebp2 expression reduces the ability to synthesize cholesterol, a major component of the myelin sheath and synaptic vesicles.

## Chloramphenicol (CAP) suppresses nSrebp2 protein expression and cholesterol synthase gene expression

Mitochondria are descendants of $\alpha$-proteobacteria, and therefore antibiotics that affect ribosomes inhibit mitochondrial translation (van den Bogert et al, 1986). To mimic a mouse model of mitochondrial translation inhibition, we treated human neuroblastoma (SH-SY5Y) cells, a neuronal cell line, with CAP, a mitochondrial translation inhibitor, and investigated whether the expression of

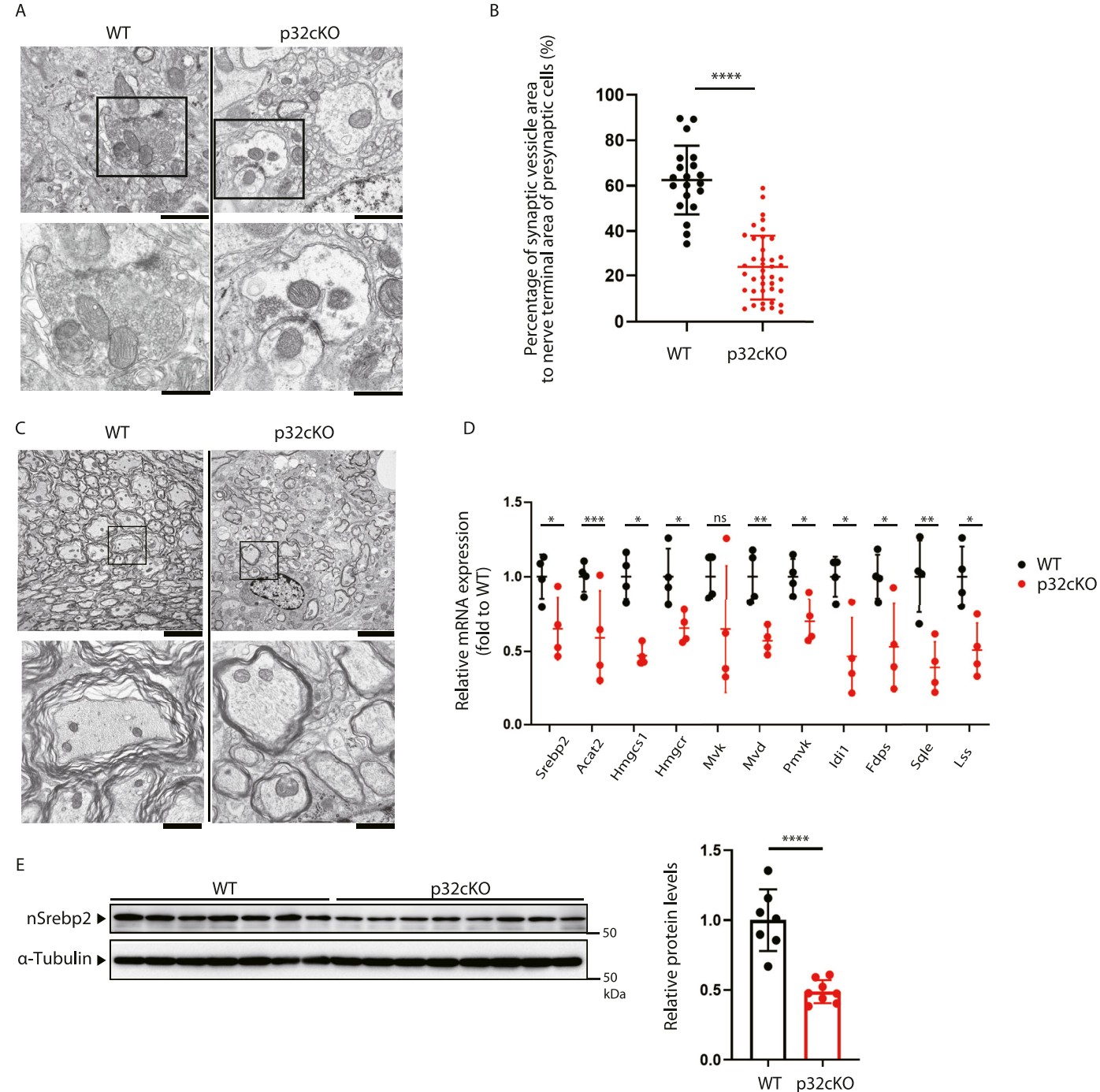

**Figure 1. Reduced synaptogenesis and decreased cholesterol synthesis gene expression in p32cKO brains.**
**(A)** The white matter region of pons and medulla oblongata from 5-wk-old wild-type and p32cKO mice were photographed using transmission electron microscopy. Mitochondria and synaptic vesicles can be seen in the neurons. Scale bars = 1 μm (upper) and 0.5 μm (lower). **(B)** The percentage of synaptic vesicles in the area of nerve endings in areas where synapses were identified in the electron microscopic images (WT: 20 locations, KO: 40 locations, n = 1 mouse per group) was calculated. Error bars represent mean ± SD. t test was performed to compare WT versus p32cKO, ****P < 0.0001. **(C)** Electron microscopic images of white matter region of pons and medulla oblongata in wild-type and p32cKO mice at 5 wk of age. p32cKO mice showed thin myelinated or unmyelinated axons in the white matter region, and mitochondria in expanded form with compact cristae. Bars = 5 μm (upper) and 1 μm (lower). **(D)** Expression of genes encoding cholesterol-synthesizing enzymes in the brains of white matter regions of 5-wk-old p32cKO mice and WT mice was analyzed by real-time PCR. Values relative to the WT were determined after correction using 18S as an internal control. Error bars represent mean ± SD. t test was performed to compare WT mice versus p32cKO mice (n = 4), *P < 0.05, **P < 0.01, ***P < 0.001. **(E)** Expression of nuclear Srebp2 (nSrebp2) in the brains of white matter regions of pons and medulla oblongata of 6-wk-old p32cKO mice and WT mice was analyzed by Western blotting. α-Tubulin expression level was used as a control. The expression of nSrebp2 was also quantified and visualized on a graph. Both values were corrected for the results of α-tubulin quantification. Error bars represent mean ± SD. t test was performed to compare WT (n = 7) versus p32cKO (n = 8), ****P < 0.0001.
Source data are available for this figure.

the cholesterol synthase genes was decreased. In undifferentiated SH-SY5Y cells treated with CAP, the expression of genes encoding a group of enzymes involved in cholesterol synthesis was reduced to about 40–50% upon 48 and 72 h of treatment compared with the level in untreated cells and 25% reduction in gene expression of the transcription factor *Srebp2* was observed (Fig 2A). In 3T3-L1 cells and MEFs treated with CAP, expression of the above-mentioned genes was reduced to about 50–60% in the treatment for 48 h and to about 10–20% in the treatment for 72 h (Fig S2A and B).

We investigated whether inhibition of mitochondrial translation similarly reduced cholesterol synthesis gene expression in differentiated SH-SY5Y cells. SH-SY5Y cells differentiate within 6 d after retinoic acid and 12-O-tetradecanoylphorbol-13-acetate (TPA) treatment, as confirmed by morphological changes and gene expression such as neuronal markers neuron-specific enolase (NSE) and tyrosine hydroxylase (TH) (Fig S2C and D). It was also observed that the expression of genes encoding a group of enzymes involved in cholesterol synthesis was reduced in differentiated SH-SY5Y cells (Fig 2B) and Oligodendrocyte cells (Fig S2E). These results suggest that the inhibition of mitochondrial translation reduces the expression of cholesterol synthase genes in neural cells and Oligodendrocyte cells.

Next, we investigated whether the decreased expression of the sterol transcription factor nSrebp2 is related to the decreased expression of the cholesterol synthase genes. For this purpose, CAP was added to SH-SY5Y cells, and the protein expression of nSrebp2 over time was measured. The absolute level of the transcription factor nSrebp2 in undifferentiated and differentiated SH-SY5Y cells decreased after 24 h of treatment with CAP (Fig 2C and D). In 3T3-L1 cells, it decreased after 60 h of treatment with CAP (Fig S2F), and in MEF cells, it decreased after 48 h of treatment with CAP (Fig S2G). We also observed that nSrebp2 was decreased by CAP treatment in Oligodendrocytes (Fig S2H).

Immunohistochemical staining was also performed to observe the expression and localisation of Srebp2 after CAP treatment. The results showed that Srebp2 was mainly present in the nucleus and that CAP treatment reduced the nuclear staining of Srebp2 (Figs 2E and S2I and J). The synthesis of nSrebp2 requires the sterol sensor Scap, and CAP treatment did not alter the expression of Scap (Fig S2K). Calnexin, a marker of the endoplasmic reticulum, was observed in the cytoplasm, indicating that Srebp2 is mainly localized to the nucleus (Figs 2E and S2L). These results suggest that the inhibition of mitochondrial translation reduces nuclear localized Srebp2 in neuron and oligodendrocytes.

## The reduced expression of the cholesterol synthase gene is because of the degradation of nSrebp2

Next, we investigated whether the decreased expression of nSrebp2 was associated with reduced cholesterol expression. For this purpose, siRNA was used to knock down Srebp2 in SH-SY5Y cells. The efficacy of Srebp2 siRNA in SH-SY5Y cells was first confirmed (Fig S3A). The expression of the cholesterol synthesis genes was reduced to the same level after 72 h of siRNA treatment as after 48 h of CAP treatment (Fig 3A). These results suggest that mitochondrial translation inhibition reduced the expression of cholesterol synthase genes by decreasing nSrebp2 expression.

Next, if down-regulation of Srebp2 by CAP suppressed cholesterol gene expression, we investigated whether overexpression of nSrebp2 in CAP-treated cells would restore cholesterol synthase gene expression. First, a nuclear transfer type nSrebp2 expression vector was generated and transfected into SH-SY5Y cells, and cholesterol gene expression was examined with and without CAP treatment. Overexpression of nSrebp2 improved cholesterol gene expression, which was suppressed by CAP, indicating that Srebp2 is a master regulator of cholesterol gene expression (Figs S3B and 3B).

Srebp2 is regulated at the transcriptional and post-translational levels, and specific signaling pathways may be involved in this regulation. The decrease in Srebp2 protein was greater than the decrease in *Srebp2* mRNA (Fig 1D and E), so we investigated whether Srebp2 is regulated at the post-translational level. The transcription factors Srebp1 and Srebp2 are ubiquitinated in the nucleus and rapidly degraded by the proteasome (Hirano et al, 2001). We hypothesized that nSrebp2 is similarly degraded by proteasomes. One study has reported that nSrebp2 is degraded by Fbw7 (F-box and WD repeat domain-containing 7) through polyubiquitination (Sundqvist et al, 2005). Therefore, we examined how cholesterol synthase expression under mitochondrial translation inhibition is rescued by the knockdown of Fbw7. We found that the knockdown of Fbw7 restored the expression of the cholesterol synthase genes that was decreased by CAP treatment (Fig 3C). Furthermore, nSrebp2 protein expression was also restored by the knockdown of Fbw7 (Fig 3D). We showed that, when a proteasome inhibitor (MG-132) was added to CAP-treated cells, nSrebp2 expression, which was reduced by CAP treatment, was restored (Fig S3C). These results suggest that, upon the inhibition of mitochondrial translation, nSrebp2 is ubiquitinated by Fbw7 and degraded by the proteasome, resulting in decreased expression of cholesterol synthase genes.

## Mitochondrial translation deficiency activates Gsk3β

Next, we focused on Gsk3β to investigate how nSrebp2 is ubiquitinated and degraded under mitochondrial translation inhibition. The activity of Gsk3β is regulated by phosphorylation, and the phosphorylation of Tyr216 in the activation loop is essential for activation (Dajani et al, 2001). It was also reported that the activation of Gsk3β phosphorylates a specific site on Srebp1 and suppresses its function (Bengoechea-Alonso & Ericsson, 2009; Dong et al, 2015). Therefore, we examined the protein expression of Gsk3β in the medulla oblongata, a region of the brain white matter, of p32cKO mice and wild-type mice. The results showed that the rate of phosphorylation of Gsk3β protein at Tyr216 (p-Gsk3β) among Gsk3β proteins was significantly increased in the brains of p32cKO mice (Fig 4A).

To confirm that similar results could be obtained in vitro, we added CAP to SH-SY5Y cells and Oligodendrocyte and examined the protein expression of Gsk3β phosphorylated at Tyr216 over time. The results showed that the phosphorylation of Gsk3β increased over time up to 6 h after CAP treatment in undifferentiated, differentiated SH-SY5Y and Oligodendrocyte (Figs 4B and C and S4A). The expression of Tyr216 phosphorylated Gsk3β protein (p-Gsk3β) was significantly increased, but we investigated whether Gsk3β activity was actually increased in the brains of p32cKO mice. These results showed that the enzymatic activity of Gsk3β was about four times higher in the white matter region of the brain of the p32cKO

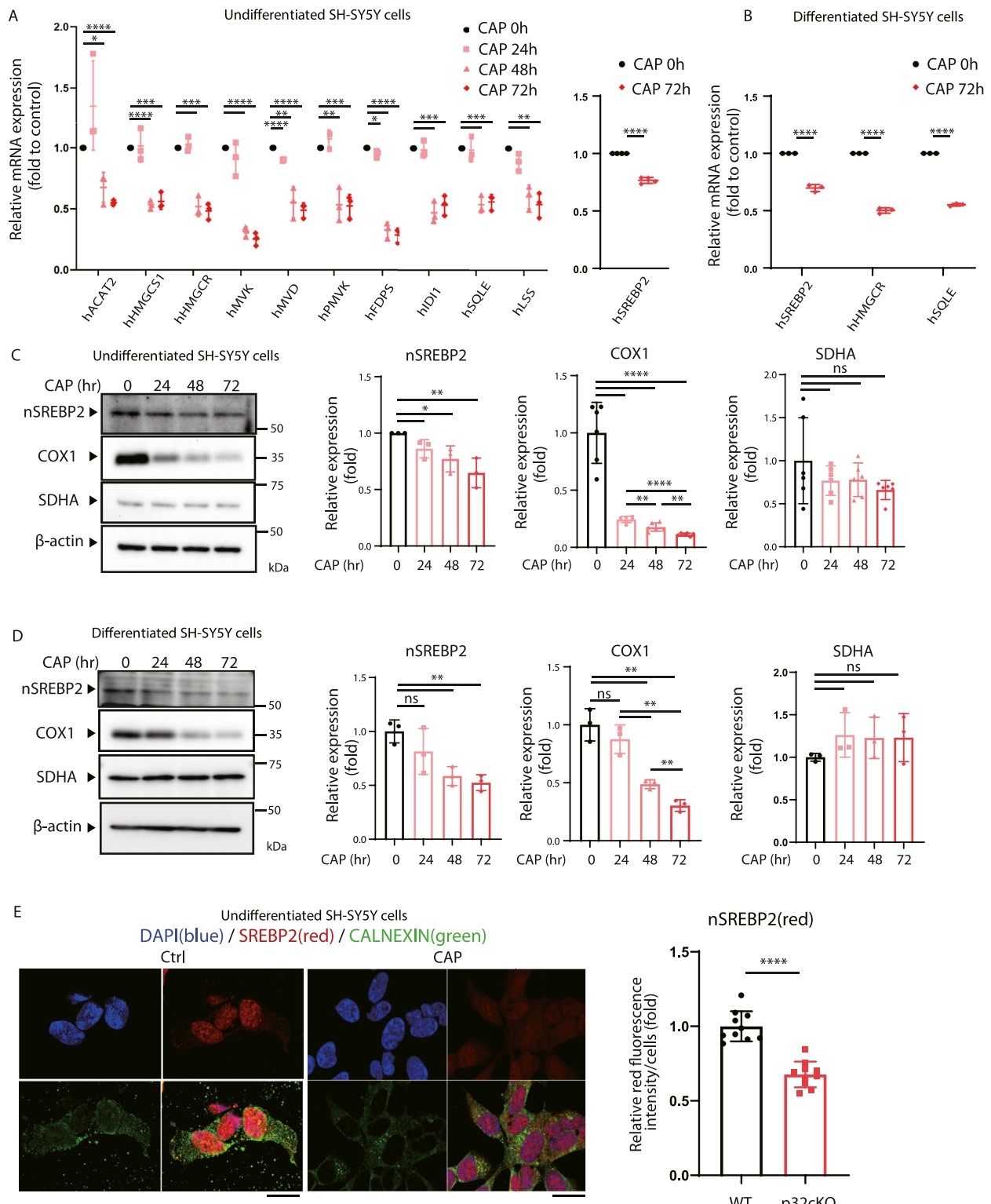

**Figure 2. CAP suppresses nSrebp2 expression and gene expression of cholesterol synthase.**

**(A)** Undifferentiated SH-SY5Y cells were treated with 100 µg/ml CAP for the indicated times, and changes in gene expression of cholesterol synthase were analyzed by real-time PCR. The expression level of 18S rRNA was measured as a control. Error bars represent mean ± SD of three independent experiments. t test was performed to compare WT cells versus WT cells treated with CAP, *P < 0.05, **P < 0.01, ***P < 0.001, ****P < 0.0001. **(B)** Differentiated SH-SY5Y cells were treated with 100 µg/ml CAP for the indicated times, and changes in gene expression of cholesterol synthase were analyzed by real-time PCR. A triple experience was carried out. **(C)** CAP (100 µg/ml) was added to undifferentiated SH-SY5Y cells and protein expression of nSrebp2 over time was analyzed by the Western blotting. The expression level of COXI which encoded mtDNA was used as positive control. SDHA which encoded nuclear DNA was used as negative control. The expression level of β-actin was used as a control. A triple

mice (Fig 4D). To determine whether activation of Gsk3$\beta$ actually suppresses Srebp2 function, we added inhibitors of GSK3$\beta$ to CAP-treated SH-SY5Y cells and examined whether they suppressed changes in cholesterol synthase gene expression. The effectiveness of GSK3 inhibitors was confirmed by phosphorylation of Gsk3$\beta$ (Ser9) (Fig S4B). Cholesterol synthase gene expression reduced by CAP was shown to be restored by treatment with GSK3$\beta$ inhibitor (Fig 4E). The expression level of nSrebp2 and nSrebp1 which was reduced by CAP treatment was also restored by GSK3$\beta$ inhibitor (Figs 4F and S4C). These results suggest that phosphorylation by Gsk3$\beta$ is involved in the down-regulation of nSrebp2 expression upon mitochondrial translation inhibition.

### Mitochondrial translation inhibition involves Pyk2-Gsk3$\beta$ activation

Several proteins have been identified as tyrosine kinases that activate Gsk3$\beta$ (Theos et al, 2005; Mccubrey et al, 2014). We focused on Pyk2, a member of the Focal Adhesion Kinase family of non-receptor tyrosine kinases, which is Ca$^{2+}$-dependent and highly expressed in neurons (Lev et al, 1995). To investigate changes in Pyk2 expression under mitochondrial translation inhibition, we first examined the phosphorylation of Pyk2 protein expressed in the medulla oblongata, a region of white matter in the brain, of p32cKO and wild-type mice (Fig 5A). We found that the expression of Pyk2 phosphorylated at Tyr402, indicating activation, was significantly increased in the white matter region of p32cKO mice. We also confirmed that the mitochondrial translation inhibitor CAP induced the expression of Pyk2 protein phosphorylated at Tyr402 in un-differentiated and differentiated SH-SY5Y cells and Oligodendrocyte (Figs 5B and C and S5A). Specifically, phosphorylated Pyk2 expression was observed after 0.5 h of CAP treatment (Fig 5B). We next investigated whether Pyk2 activates Gsk3$\beta$ using Pyk2 inhibitors. Phosphorylated Gsk3$\beta$ (Tyr216) expression, increased by CAP, was suppressed by Pyk2 inhibitors (Fig 5D). However, GSK3$\beta$ inhibitor did not inhibit the phosphorylation of Pyk2, suggesting that Pyk2 is a kinase upstream of Gsk3$\beta$ (Fig S5B). In SH-SY5Y, Pyk2 inhibition was also observed to restore nSrebp2 expression by CAP treatment (Fig 5E). These results indicate that phosphorylated Pyk2, the activity of which is increased upon mitochondrial translation inhibition, activates Gsk3$\beta$, leading to the suppression of nSrebp2 function.

## Discussion

In mice with neuron-specific conditional knockout of p32, which is involved in mitochondrial translation, demyelination was observed in the brain, leading to leukoencephalopathy and death after only 8 wk (Yagi et al, 2017). Clarifying the mechanism behind this is

important because it could lead to the establishment of effective treatments of demyelination-related diseases. We focused on cholesterol, a major component of myelin sheaths and synaptic vesicles, and found that inhibition of mitochondrial translation suppresses the function of the sterol regulator Srebp2, leading to reduced expression of cholesterol synthesis genes. It is considered that suppression of cholesterol biosynthesis can lead to demyelination. Decreased Srebp2 is associated with reduced Srebp2 mRNA expression and increased degradation of Srebp2 protein. In this study, we analyzed in detail the mechanism of Srebp2 protein degradation and found that the Pyk2-GSK3 axis is involved.

The Srebp1 and Srebp2 have a bHLH-Zip region and ~45% amino acid homology. In both of these proteins, the N-terminal side translocates to the nucleus and exerts transcription factor activity, and Srebp2 activates genes mainly encoding cholesterol and LDL receptors (Guo et al, 2024). Srebp2 is regulated by post-translational modifications and it has been reported that nSrebp2 is degraded by the ubiquitin-proteasome system as a contributing factor to loss of function. In this study, we demonstrated that ubiquitination by the ubiquitin ligase SCF$^{Fbw7}$ is responsible for the loss of nSrebp2 function in mitochondrial translation inhibition, suggesting that degradation of Srebp2 may affect the amount of Srebp2 protein and reduce cholesterol gene expression.

Cholesterol is also a fundamental component of synaptic vesicles, which account for neurotransmission at nerve endings. The membranes of synaptic vesicles are also enriched in cholesterol, the abundance of which exceeds that in the plasma membrane (Breckenridge et al, 1973). Cholesterol has also been reported to promote the membrane fusion of synaptic vesicles by interacting with synaptophysin (Thiele et al, 2000). Decreased cholesterol metabolism is observed in the brain during neurodegenerative diseases such as Alzheimer's disease, Parkinson's disease, Huntington's disease, and amyotrophic lateral sclerosis but also during aging (Djelti et al, 2015; Paul et al, 2017). In Alzheimer's disease, brain cholesterol levels are markedly reduced compared with those in healthy brains (Ledesma et al, 2003), suggesting that demyelination of the brain's white matter leads to a loss of capacitance and impaired nerve conduction, which can severely impair brain function. In addition, the most significant risk factor for developing Alzheimer's disease is aging, with the frequency of Alzheimer's disease rising substantially with age (2021 Alzheimer's Disease Facts and Figures, 2021). These findings implicate age-related mitochondrial dysfunction in the demyelination observed in neurodegenerative diseases.

Gsk3$\beta$, which is associated with neurodegenerative diseases, including Alzheimer's disease, is predominantly expressed in the central nervous system and has been reported to phosphorylate SREBP-1c and promote its degradation upon activation (Dong et al, 2015). In this study, we found that Gsk3$\beta$ was involved in the loss of

---

experience was carried out. **(D)** CAP (100 μg/ml) was added to differentiated SH-SY5Y cells and protein expression of nSrebp2 over time was analyzed by the Western blotting. Triplicated experience was performed. **(E)** Immunofluorescence experience with anti-calnexin antibody (green), anti-Srebp2 antibody (red) and DAPI (blue) were performed in SH-SY5Y cells treated with CAP. The lower right panels show merged images. Scale bar = 10 μm. The right panel shows quantification of nSrebp2 fluorescence intensity in the nucleus (n = 10).

Source data are available for this figure.

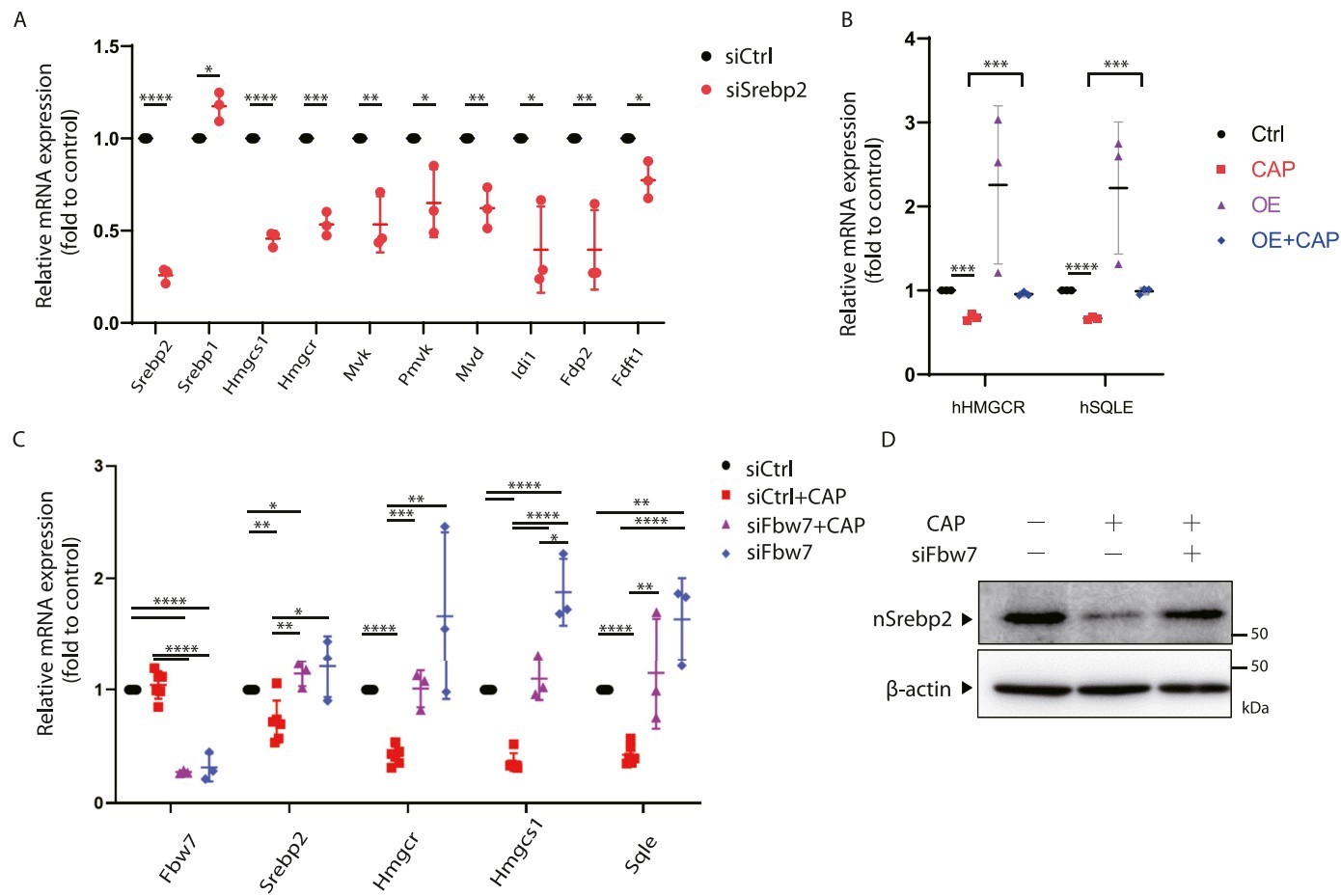

**Figure 3. Reduced Srebp2 involved in cholesterol gene expression.**
**(A)** MEF cells were transfected with 25 pmol siSrebp2 and cells were collected 72 h later. Changes in the expression of genes encoding cholesterol synthesis system enzymes by the knockdown of Srebp2 were evaluated by real-time PCR. The expression level of 18S rRNA was measured as a control. Error bars represent mean ± SD of three independent experiments. *t* test was performed to compare WT cells versus WT cells transfected with siSrebp2, *$P < 0.05$, **$P < 0.01$, ***$P < 0.001$, ****$P < 0.0001$. **(B)** Overexpression of nSrebp2 in SH-SY5Y cells prevents the down regulation of cholesterol gene expression induced by chloramphenicol. A triple experience was carried out. **(C)** siRNA (25 pmol) was transfected into $1 \times 10^5$ MEF cells/well to knock down Fbw7. 72 h later, cells were collected and changes in expression of cholesterol synthase genes because of Fbw7 knockdown were evaluated by real-time PCR. Chloramphenicol (100 µg/ml) treatment was performed for 24 h. The expression level of 18S rRNA was measured as a control. Error bars represent mean ± SD of three or six independent experiments. **(D)** siRNA (25 pmol) was transfected into MEF cells to knock down Fbw7. 72 h later, cells were collected and analyzed for nSrebp2 expression by the Western blotting (n = 1). The expression level of β-actin was used as a control. Source data are available for this figure.

Srebp2 function, suggesting that Gsk3β is responsible for the functional regulation of Srebp2. The phosphorylation sites were not identified in this study, but as previous studies have reported that the phosphorylation sites of Srebp1a are Thr-426 and Ser-430, (Bengoechea-Alonso & Ericsson, 2009). We speculated that the phosphorylation sites of Srebp2 are Ser-432 and Ser-436; further functional analysis is required to identify the phosphorylation sites in the future.

Various factors have been reported to activate Gsk3β, including Akt (Wakatsuki et al, 2011), Zak1 (Kim et al, 1999), and Fyn (Lesort et al, 1999). Because Tyr216 of Gsk3β is phosphorylated upon mitochondrial translation inhibition, we focused on the tyrosine kinase Pyk2, which is involved in the pathogenic mechanism of neurodegenerative diseases such as Alzheimer's disease. In eukaryotic cells, there is close interaction between mitochondria and endoplasmic reticulum, forming mitochondria-associated

endoplasmic reticulum membranes (MAMs), which are physically and biochemically linked (Marchi et al, 2014). Enlarged mitochondria with an abnormal internal structure have been observed in the hearts of p32-deficient mice (Saito et al, 2017), and similarly swollen mitochondria have been observed in the brains of mice with neuron-specific p32cKO (Yagi et al, 2017). MAMs function as a site of regulation of intracellular lipid, cholesterol, and calcium homeostasis, and increased coupling between mitochondria and the endoplasmic reticulum has been reported to result in abnormal cholesterol homeostasis (Stefani & Liguri, 2009) and increased calcium transport (Bezprozvanny & Mattson, 2008).

In neurons, Pyk2 is phosphorylated upon $Ca^{2+}$ influx and partially translocated to mitochondria (Hirschler-Laszkiewicz et al, 2018; Miller et al, 2019); in addition, Pyk2 has been reported to localize to MAMs involved in calcium homeostasis (López-Molina et al, 2022). These observations suggest that the inhibition of mitochondrial

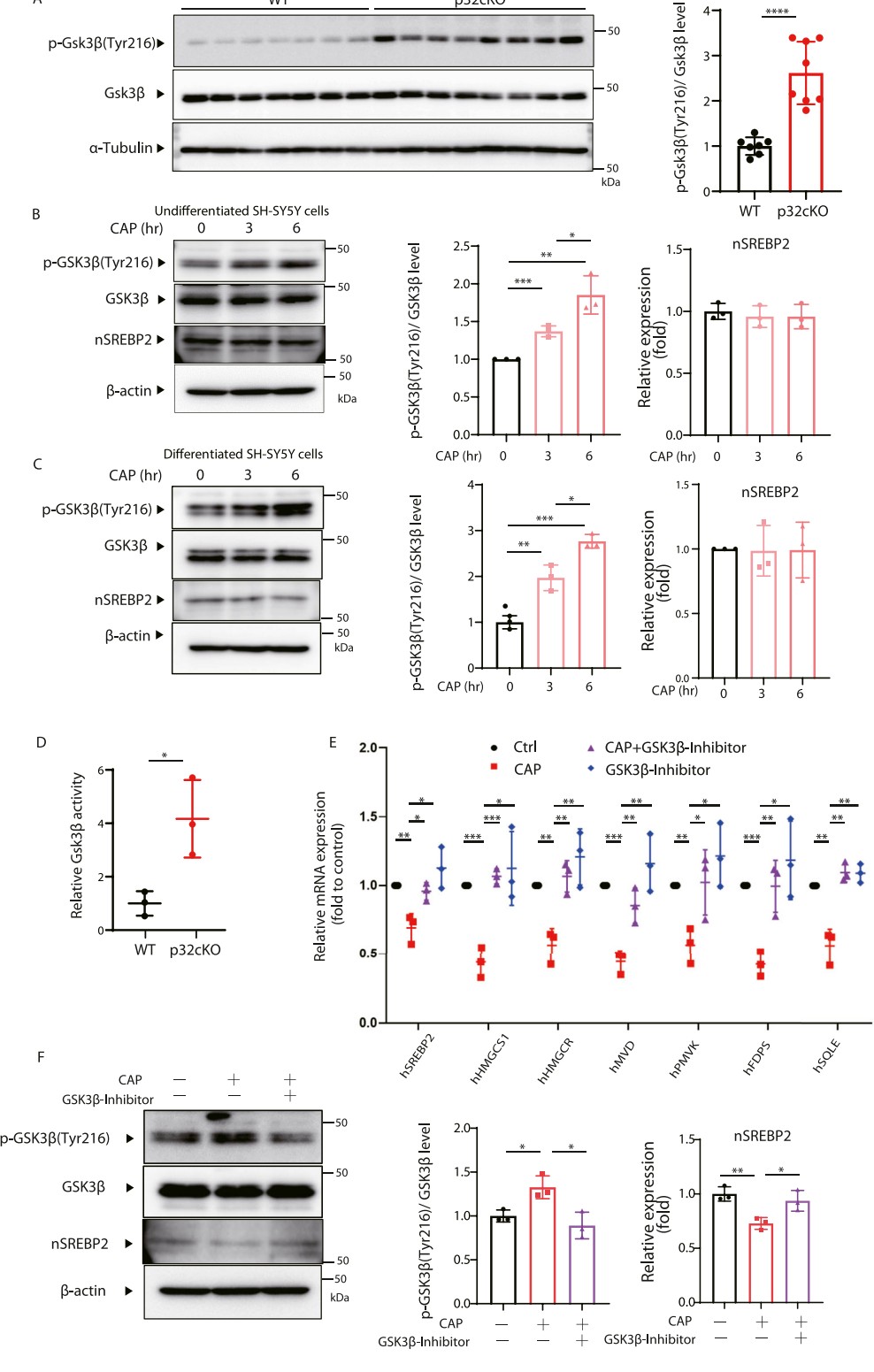

**Figure 4. Inhibition of mitochondrial translation activates Gsk3β.**
(A) Expression of p-Gsk3β (Tyr216) and Gsk3β in the brains of 6-wk-old p32cKO mice and wild-type mice was analyzed by the WB method. The expression level of α-tubulin was used as a control. In addition, the rate of phosphorylation of Tyr216 in Gsk3β was quantified and visualized graphically. In both cases, values were corrected using the results of α-tubulin quantification. Error bars represent mean ± SD. t test was performed to compare WT (n = 7) versus p32cKO (n = 8), ****P < 0.0001. (B) CAP (100 μg/ml) was added to undifferentiated SH-SY5Y cells, and the phosphorylation of Gsk3β at Tyr216 over time and changes in Gsk3β were analyzed by the Western blotting. nSrebp2 and β-actin expression level was used as a control. The right panel shows quantification the phosphorylation rate of Tyr216 in GSK3β. Triplicated experience was performed. t test was performed to compare WT cells versus WT cells treated with CAP, *P < 0.05, **P < 0.01, ***P < 0.001. (C) CAP (100 μg/ml) was added to differentiated SH-SY5Y cells, and the phosphorylation of Gsk3β at Tyr216 over time and changes in Gsk3β were analyzed by the Western blotting. nSrebp2 and β-actin expression level was used as a control. The right panel shows quantification the phosphorylation rate of Tyr216 in Gsk3β. Triplicated experience was performed. (D) Gsk3β activity levels were measured in the brains of 6-wk-old p32cKO mice and wild-type mice. Error bars represent mean ± SD. t test was performed to compare WT (n = 3) versus p32cKO (n = 3), *P < 0.05. (E) Expression of cholesterol synthase genes in SH-SY5Y cells treated with CAP (100 μg/ml) or GSK3β inhibitor (tideglusib) (20 μM), or left untreated was analyzed using real-time PCR. The treatment with GSK3β inhibitor and CAP was performed for 48 h. The expression level of 18S rRNA was measured as a control. Error bars represent mean ± SD of three independent experiments. (F) Proteins were extracted from SH-SY5Y cells treated with CAP (100 μg/ml) or GSK3β inhibitor (CHIR-99021) (20 μM), or left untreated and analyzed for nSrebp2 expression using the Western blotting. β-Actin expression level was used as a control. A triple experience was carried out. The right panel shows quantification the phosphorylation rate of Tyr216 in Gsk3β. Source data are available for this figure.

translation may lead to mitochondrial expansion, which enhances calcium transport in MAMs and activates Pyk2. This suggests that the inhibition of mitochondrial translation function leads to the disruption of calcium homeostasis in neurons. In addition to senile plaques and neurofibrillary changes, the pathophysiology of Alzheimer's disease includes abnormal cholesterol metabolism,

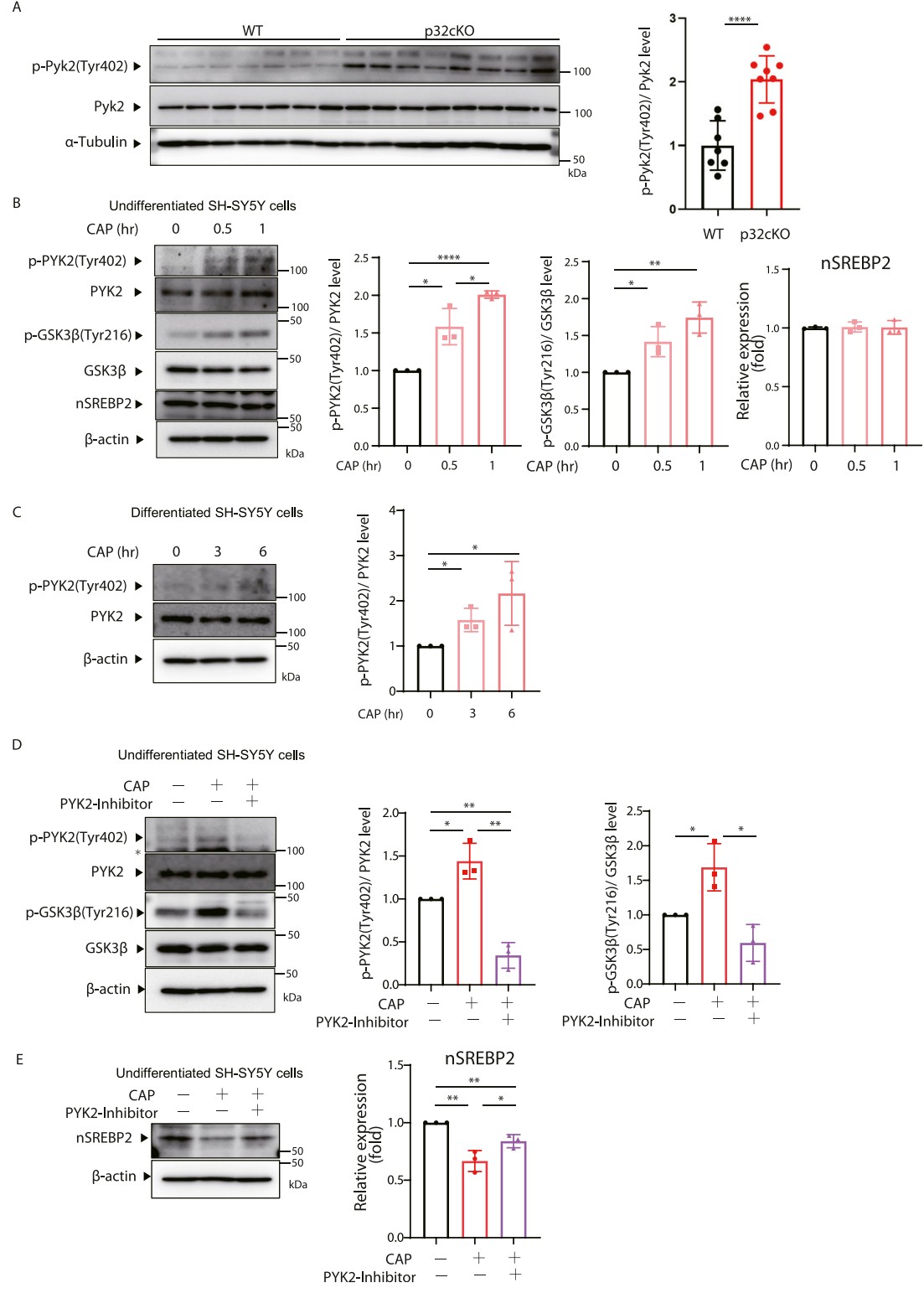

**Figure 5. Inhibition of mitochondrial translation activates Pyk2, which contributes to Gsk3β activation.**
**(A)** The expression of p-Pyk2 (Tyr402) and Pyk2 in the brains of 6-wk-old p32cKO mice (n = 8) and wild-type mice (n = 7) was analyzed by the Western blotting. The expression level of α-tubulin was used as a control. The rate of phosphorylation of Tyr402 in Pyk2 was quantified and visualized graphically. Both values were corrected using the results of α-tubulin quantification. Error bars represent mean ± SD. t test was performed to compare WT (n = 7) versus p32cKO (n = 8), ****P < 0.0001. **(B)** CAP (100 μg/ml) was added to undifferentiated SH-SY5Y cells, and changes in Pyk2 phosphorylation of Tyr402, Pyk2, Gsk3β phosphorylation of Tyr216, and Gsk3β protein expression over time were analyzed by the Western blotting. The expression level of β-actin was used as a control. A triple experience was carried out. The right panel shows quantification the phosphorylation rate of Tyr402 in Pyk2 and Tyr216 in Gsk3β. **(C)** CAP (100 μg/ml) was added to differentiated SH-SY5Y cells, and changes in Pyk2

abnormal phospholipid metabolism, abnormal calcium homeostasis, and abnormal mitochondrial function, all of which are functional abnormalities associated with MAMs (Area-Gomez & Schon, 2016; Tambini et al, 2016).

Another gene whose expression is regulated by Srebp2 is *low-density-lipoprotein receptor* (*Ldlr*) (Nagoshi et al, 1999). Oligodendrocytes synthesize cholesterol intracellularly, whereas some neurons can take up cholesterol from astrocytes. There are receptors for LDL, an apolipoprotein, on neuronal membranes; one member of the Ldlr family is ApoER (Apolipoprotein E Receptor), which binds to ApoE in LDL, thereby transporting cholesterol to neurons. The production of cholesterol required for synaptogenesis and neuronal repair depends on this ApoE–ApoER pathway and Ldlr gene expression was found to be reduced by the inhibition of mitochondrial translation by CAP. Thus, the demyelination observed in the mice with neuron-specific p32cKO may be because of decreased synthesis of cholesterol by the Pyk2–Gsk3$\beta$–Srebp2 pathway, as well as a decreased ability to incorporate cholesterol into the cytosol, which is required for membrane synthesis using extracellular sources.

The newly revealed mechanism of reduced cholesterol synthesis because of the suppression of Srebp2 function associated with the activation of Pyk2–Gsk3$\beta$ by mitochondrial translation defects supports the MAM hypothesis and is expected to help in elucidating the molecular mechanisms behind neurodegenerative diseases caused by aging.

# Materials and Methods

### Mating of knockout mice

The protocol for the experimental animal studies was approved by Kyushu University Animal Ethics Committee (approval No. #A21-182-1, A23-144-0). All animal experimental protocols adhered to the Guidelines of the Care and Use of Laboratory Animals, Eighth Edition, updated by the US National Research Council Committee in 2011. The current study also adheres to the ARRIVE Guidelines for reporting in vivo experiments. Mice were kept in a light/dark cycle at 22°C for 12 h. Distilled water and solid feed for rodents were consumed ad libitum. p32$^{flox/flox}$ mice with the same genetic background of the C57BL/6 strain and Nestin-Cre transgenic mice [B6.Cg-Tg(Nestin-cre)1Kln/J] (The Jackson Laboratory) were crossed (Yagi et al, 2012). Genotyping of the p32 and Cre alleles was performed by PCR using DNA extracted from the tails of the mice (Yagi et al, 2012).

### Electron microscopy

A JEOL 1200 electron microscope (JEOL) was used for transmission electron microscopy. Qualitative evaluation of the observed neural structures and detailed histomorphometric analysis were performed based on previous studies (Amamoto et al, 2011). First, mouse brain was perfusion-fixed using a fixative solution diluted to 2% paraformaldehyde (GA) in 0.1 M cacodylate buffer (pH 7.4). Then, the samples were fixed in 0.1 M cacodylate buffer (pH 7.4) containing 2% PFA, 2% GA, and 0.5% tannic acid for 2 h at 40°C. The samples were next washed four times for 15 min each with 0.1 M cacodylate buffer and post-fixed in 2% osmium tetroxide (OsO$_4$) diluted in 0.1 M cacodylate buffer at 40°C for 2 h. The samples were subsequently dehydrated using a gradated ethanol series (50%, 70%: 30 min at 40°C, 90%: 30 min at room temperature, 100%: 30 min at room temperature). They were then infiltrated with propylene oxide (PO) twice for 30 min each and sealed in a mixture of PO and embedding resin (Quetol-812; Nissin EM Co., Ltd.) at a ratio of 7:3 for 1 h. The lid was then opened and the PO was allowed to volatilize overnight. The samples were transferred to the encapsulated resin and allowed to polymerize at 60°C for 48 h. The resin blocks underwent quasi-ultrathin sectioning to 1.5 $\mu$m with a glass knife using an Ultramicrotome Ultracut-UCT (Leica) and were stained with 0.5% toluidine blue. Ultrathin sectioning was then performed on the blocks at 70 nm with a diamond knife on an Ultramicrotome Ultracut-UCT (Leica). Copper grids were used as holders. The sections were then stained with 2% aqueous uranyl acetate solution for 15 min at room temperature, rinsed with distilled water, and further stained with lead citrate staining solution (Sigma-Aldrich) for 3 min at room temperature. Digital images of 2048 × 2048 pixels were taken with a VELETA (Olympus Soft Imaging Solutions GmbH). ImageJ (Schneider et al, 2012) was used to calculate the area of synaptic vesicles.

### RNA extraction from brain tissue

Tissues were harvested from the brains of 5-wk-old mice with neuron-specific p32 knockout mice, immersed in RNAlater Solution (Invitrogen), and cryopreserved. To a 20 mg piece of tissue was added 500 $\mu$l of Lysis Buffer (LBA buffer with 2% 1-thioglycerol) and the tissue was disrupted by pipetting. Equal volumes of Lysis Buffer and RNA Dilution Buffer were added, followed by vortexing for 10 s. The cells were left at room temperature for 1 min and centrifuged at 14,000$g$ for 3 min at room temperature. In addition, 500 $\mu$l of isopropanol was added, followed by vortexing for 5 s. The lysate was then added to a minicolumn set in a collection tube and centrifuged at 14,000$g$ for 1 min at room temperature. Subsequently, 500 $\mu$l of RNA Wash Solution was added to the column and

---

phosphorylation of Tyr402 and Pyk2 protein expression over time were analyzed by the Western blotting. The expression level of $\beta$-actin was used as a control. A triple experience was carried out. The right panel shows quantification the phosphorylation rate of Tyr402 in Pyk2. **(D)** Proteins were extracted from SH-SY5Y cells treated with CAP (100 $\mu$g/ml) for 3 h or PYK2 inhibitor (PF-562271) (100 $\mu$M) for 3 h, or left untreated and analyzed for p-Gsk3$\beta$, Gsk3$\beta$, p-Pyk2, and Pyk2 expression using the WB method. $\beta$-Actin expression level was used as a control. A triple experience was carried out. The right panel shows quantification the phosphorylation rate of Tyr402 in Pyk2 and Tyr216 in Gsk3$\beta$. * showed non-specific band. **(E)** Proteins were extracted from SH-SY5Y cells treated with CAP (100 $\mu$g/ml) for 72 h or PYK2 inhibitor (PF-562271) (1 $\mu$M) for 72 h, or left untreated and analyzed for nSrebp2 expression using the Western blotting. $\beta$-Actin expression level was used as a control. A triple experience was carried out. The right panel shows the quantification of nSrebp2.
Source data are available for this figure.

centrifuged at 14,000$g$ for 30 s at room temperature. Next, 30 $\mu$l of DNase Incubation Mix (24 $\mu$l of Yellow Core Buffer, 3 $\mu$l of MnCl$_2$ [0.09%], 3 $\mu$l of DNaseI Solution) was added to the minicolumn, which was then left to stand at room temperature for 15 min. The minicolumn was centrifuged at 14,000$g$ for 15 s at room temperature. The column was again washed twice with RNA Wash Solution and extracted with nuclease-free water.

### Real-time PCR analysis

Total RNA was extracted from the recovered cells using the RNeasy Mini Kit (74104; Qiagen), and the total RNA was treated with DNase I (100 U) of the RNase-Free DNase Set (79254; Qiagen) for 15 min at room temperature. RNA was cleaned up using the RNeasy Mini Kit. The RNA concentration was quantified with NanoDrop (ND-1000) (NanoDrop Technologies), and 500 ng of total RNA was reverse-transcribed with PrimeScript RT Master Mix (Perfect Real Time) (RR037A; Takara) to synthesize cDNA. The PCR enzyme used was TB Green Premix Ex Taq II (Tli RNaseH Plus) (RR820A; Takara), and quantitative PCR was performed using the StepOnePlus Real-Time PCR System (Applied Biosystems). The PCR primers used are listed in Table S1.

### Protein extraction from brain tissue

Brain tissue was removed from 5-wk-old mice with neuron-specific knockout of p32, cut into small pieces, placed in tubes, and flash-frozen in liquid nitrogen. For protein extraction, Lysis Buffer (20 mM Tris pH7.5, 150 mM NaCl, 2 mM EDTA, 1% NP40) containing Protease Inhibitor Cocktail (169-26063; Wako) and PhosSTOP (04-906-837-001; Roche) was used. The supernatant obtained by centrifugation at 20,000$g$ for 5 min at 4°C was further recentrifuged under the same conditions to purify the protein.

### Western blotting

Cells were solubilized in Lysis Buffer (20 mM Tris pH7.5, 150 mM NaCl, 2 mM EDTA, 1% NP40) containing Protease Inhibitor Cocktail (169-26063; Wako) and PhosSTOP (04-906-837-001; Roche). After solubilization in Lysis Buffer (20 mM Tris pH7.5, 150 mM NaCl, 2 mM EDTA, 1% NP40), the samples were homogenized by sonication. Protein concentration was measured by SPECTROstar Nano (BMG Labtech), and SDS–PAGE was performed with 5 $\mu$g of sample per lane. Running Buffer (25 mM Tris, 190 mM glycine, 0.1% SDS) was used for electrophoresis, and Transfer Buffer (25 mM Tris, 190 mM glycine, 20% methanol) was used for transcription on a polyvinylidene fluoride membrane (IPVH00010; Merck). In addition, Precision Plus Protein Dual Color Standards (161-0394; Bio-Rad) were used as protein standards. After blocking at room temperature for 2 h, primary antibodies were reacted overnight at 4°C using a solution of 5% Nonfat Dry Milk (9999S; Cell Signaling) dissolved in 1× PBST (PBS, 0.1% Tween20). The solute for the primary antibody solution was prepared by dissolving Bovine Serum Albumin (A7906-50G; Sigma-Aldrich) in 1× PBST at a concentration of 1%. We used the following primary antibodies for Western blotting: Srebp2 (10007663; Cayman Chemical), Gsk3$\beta$ (9315P; Cell Signaling), Gsk3$\beta$ [pTyr216] (NB100-81946; Novus Biologicals), Gsk3$\beta$ [pSer9] (9323; Cell

Signaling), Pyk2 (3292S; Cell Signaling), Pyk2 [pTyr402] (3291S; Cell Signaling), COX1 (459600; Novex by life technologies), SDHA (459200; Invitrogen), $\beta$-actin (#A5441; Sigma-Aldrich), $\alpha$-Tubulin (ab7291; Abcam), Scap (13102; Cell Signaling), Srebp1 (ab28481; Abcam). After that, the membrane was washed with 1× PBST and the secondary antibody was reacted for 2 h at 4°C. The solute for the secondary antibody solution was prepared by dissolving Nonfat Dry Milk in 1× PBST at a concentration of 1%. We used the following secondary antibodies for Western blotting: anti-rabbit IgG HRP-linked (#7074; CST) and anti-mouse IgG HRP-linked (#7076; CST). The detection reagents were Clarity Western ECL substrate (170-5061; Bio-Rad) or Western Lightning ECL Pro (NEL121001EA; PerkinElmer). Detection was performed using a GE ImageQuant LAS 4000 Mini (GE Healthcare).

### Cell culture

MEF cells, 3T3-L1 cells (mouse fetal fibroblasts), and SH-SY5Y cells (human neuroblastoma cells), HOG cells (Human Oligodendroglioma cells) were cultured in an incubator (SCA-165DS; ASTEC) at 5% CO$_2$ and 37°C. The culture medium consisted of non-activated 10% FBS (F7524; Sigma-Aldrich) and Penicillin–Streptomycin Mixed Solution (09367-34; Nacalai Tasque) supplemented with DMEM-low glucose (D6046; Sigma-Aldrich). For passaging, 0.5%-Trypsin/5.3 mM EDTA Solution (35556-44; Nacalai Tasque) was used. A Coulter Counter ZIS (Yamato Kagaku) was used for cell counting. Accutase (SCR005; Merck) and PBS(−) (166-23555; Wako) were also used in the culture of HOG cells.

### Mitochondrial translation inhibitor treatment

Samples of SH-SY5Y, MEF, 3T3-L1, and HOG cells used in real-time PCR and Western blotting were treated with CAP at a final concentration of 100 $\mu$g/ml. The cells were seeded in six-well plates and the drugs were added 24 h later.

### RNAi

Short interfering RNAs (siRNAs) for Silencer Select siRNA for the Srebp2 gene (siRNA ID: s74387, s74388, s74389; Cat. No. 4390771), Silencer Select predesigned siRNAs, were obtained from Ambion (Thermo Fisher Scientific). The following siRNAs were used to transfect cells: sequence 1, CAGCCUUUGAUAUACCAGATT (sense) and UCUGGUAUAUCAAAGGCUGCT (antisense); sequence 2, GCAGUACAGCGGUCAUUCATT (sense) and UGAAUGACCGCUGUACUGCAG (antisense); and sequence 3, CUGGUACGCUGGUUACUCATT (sense) and UGAGUAACCAGCGUACCAGGC (antisense). A total of 150 $\mu$l of Opti-MEM I (31985-062; Thermo Fisher Scientific) with 25 pmol siRNA and 150 $\mu$l of Opti-MEM I with Lipofectamine RNAiMAX (13778-150; Invitrogen) was incubated for 5 min at room temperature, mixed, and then incubated for another 30 min at room temperature. A total of 1 × 10$^5$ MEF or 3T3-L1 cells were added to the mixed solution and added to 1.7 ml of culture medium in a six-well dish after 5 min of incubation. Cell collection was performed 72 h after seeding, and the changes in RNA expression and protein expression were evaluated.

## Drug treatment

The activator and inhibitor used in this study for MEF, 3T3-L1, SH-SY5Y, and HOG cells were Tideglusib (SML0339; Sigma-Aldrich), CHIR-99021(HY-10182; MedChemExpress), PF-562271 (HY-10459; MedChemExpress), and MG-132 (HY-13259; MedChemExpress), respectively.

## Immunocytochemistry

A total of $1 \times 10^4$ SH-SY5Y cells were seeded in 24-well dishes lined with PLL-coated Cover Glass 12 mm type (4912-040; IWAKI), and the cells were cultured in DMEM with 10% FBS at 37°C and 5% $CO_2$. Next, 4% Paraformaldehyde Phosphate Buffer Solution (09154-85; Nacalai Tasque) was used for fixation for 10 min at room temperature. Both primary and secondary antibodies were used at 200-fold dilution. As the first antibody, Calnexin (66903; proteintech) was used. As the secondary antibody, Goat Anti-Rabbit IgG (H + L) Cross-Adsorbed Secondary Antibody, Alexa Fluor 594 (A11012; Invitrogen), was used. Fluorescence images were obtained using a fluorescence microscope (BZ-x700; KEYENCE).

## Expression constructs

An expression construct containing the SREBP2 cDNA was generated by standard methods. cDNAs of wild-type was cloned into the BamHI/EcoRI sites of the expression vector pcDNA3 (Invitrogen). A 1-468aa SREBP2 cDNA containing the deduced first methionine site was amplified from a cDNA library of SH-SY5Y cells by PCR using the primer set: 5'-ATG GAC GAC AGC GGC G-3' and 5'-GCC CTA GTC TGG CTC ATC TTT G-3'. Then, BamHI and EcoRI sites were added to the 5'-and 3'-terminals, respectively, of the cDNA by a second PCR using the primers 5'-AAA TTT GGA TCC ATG GAC GAC AGC GGC-3' and 5'-AAA TTT GAA TTC CTA GTC TGG CTC ATC TTT GA-3'. The PCR product was digested with BamHI and EcoRI.

## Gsk3β activity measurement

Brain tissues from 6-wk-old mice with neuron-specific knockout of p32 were harvested and subjected to protein purification. Protein was measured using SPECTROstar Nano (BMG Labtech) and each sample was adjusted to a final concentration of 20 ng/µl. Master Mix (5 µl of 5× kinase assay buffer, 1 µl of 500 µM ATP, 5 µl of 1 mg/ml GSK Substrate, 14 µl of MilliQ) was prepared, to which 1 µl of a 750 ng/µl sample was added, followed by the addition of 9 µl of 1× kinase assay buffer and 15 µl of MilliQ. The mixture was then incubated at 30°C for 45 min and dispensed onto a white 96-well plate (BPS Bioscience). After that, 50 µl of Kinase-GloMax Assay (Promega) was added, and the reaction was carried out for 15 min at room temperature under light-shielded conditions. Samples were then measured with Multiplate Plate Reader ARVO X2 (PerkinElmer Co.). The software PerkinElmer 2030 Workstation (PerkinElmer Co., Ltd.) was used to analyze. To subtract the effect of ATP-consuming enzymes other than Gsk3β in the samples, we simultaneously measured the Master Mix without the substrate as a negative control.

## Immunohistochemistry

Immunohistochemical staining was performed using the His-tofine SAB-PO kit (Nichirei). First, mouse cerebellum was fixed with 10% neutral buffered formalin and paraffin-embedded blocks were prepared. Then, 4-µm-thick sections were cut using a microtome, deparaffinized with xylene, and dehydrated with ethanol. The endogenous peroxidase activity was inactivated by methanol with 0.3% hydrogen peroxide for 30 min. After quenching, the sections were blocked with 10% normal rabbit serum dissolved in phosphate buffer for 10 min. The sections were then reacted with anti-synaptophysin antibody for 2 h at room temperature. Secondary antibodies were then incubated for 20 min at room temperature. DAB (3,3'-Diaminobenzidine-tetrahydrochloride) was used as a chromogenic substrate. ABZ-X700 microscope (Keyence) was used to observe the samples. Mice were anesthetized with an overdose of sevoflurane. After exsanguination under deep anesthesia, tissues were fixed in 4% paraformaldehyde and paraffin-embedded coronal sections were prepared for histological staining with Kluver–Barrera stain. Immunohistochemistry was performed as described previously (Amamoto et al, 2011).

## Quantification and statistical analysis

The data are expressed as means ± SD of the indicated number of experiment and mice. Unpaired $t$ test was used to determine statistical differences between two groups. *$P$ < 0.05, **$P$ < 0.01, ***$P$ < 0.001, or ****$P$ < 0.0001 was considered statistically significant.

# Data Availability

Further information and requests for resources and reagents should be directed to and will be made available upon reasonable request by Takeshi Uchiumi (uchiumi.takeshi.008@m.kyushu-u.ac.jp). This study did not generate new reagents.

# Supplementary Information

# Acknowledgements

We also thank Edanz (https://jp.edanz.com/english-editing-c) for editing a draft of this manuscript. We also appreciate laboratory members for reagents, discussions, and critical reading of the manuscript. This work was supported by Japan Society for the Promotion of Science (JSPS) Grant Numbers #22H03537, #23K18217, #20H00530. We are grateful for the technical support provided by the Research Support Center, Graduate School of Medical Sciences, Kyushu University.

## Author Contributions

T Toshima: Resources, data curation, formal analysis, validation, investigation, visualization, methodology, project administration, and writing—original draft, review, and editing.
M Yagi: data curation, validation, investigation, methodology, and writing—review and editing.
Y Do: data curation, validation, investigation, and writing—review and editing.
H Hirai: data curation, validation, investigation, and writing—review and editing.
Y Kunisaki: conceptualization, supervision, and writing—review and editing.
D Kang: conceptualization, supervision, funding acquisition, and writing—review and editing.
T Uchiumi: conceptualization, resources, data curation, formal analysis, supervision, funding acquisition, validation, investigation, visualization, methodology, project administration, and writing—original draft, review, and editing.

## Conflict of Interest Statement

The authors declare that they have no conflict of interest.

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
