## [Reviewer comments · Life Science Alliance]

Life Science Alliance

Mitochondrial translation failure represses cholesterol gene expression via Pyk2-Gsk3 β -Srebp2 axis

Takahiro Toshima, Mikako Yagi, Yura Do, Haruka Hirai, Yuya Kunisaki, Dongchon Kang, and Takeshi Uchiumi
DOI: <https://doi.org/10.26508/lsa.202302423>

Corresponding author(s): Takeshi Uchiumi, Kyushu University

Review Timeline:

Submission Date:	2023-10-07
Editorial Decision:	2023-11-17
Revision Received:	2024-04-21
Editorial Decision:	2024-04-25
Revision Received:	2024-04-27
Accepted:	2024-04-29

Transaction Report:

November 17, 2023

Re: Life Science Alliance manuscript #LSA-2023-02423

Dr. Takeshi Uchiumi
Kyushu University
Clinical Chemistry and Laboratory Medicine
3-1-1 Maidashi
Fukuoka, Fukuoka 812-8582
Japan

Dear Dr. Uchiumi,

Thank you for submitting your manuscript entitled "Mitochondrial translation failure represses cholesterol gene expression via Pyk2-Gsk3 β -Srebp2 axis" to Life Science Alliance. The manuscript was assessed by expert reviewers, whose comments are appended to this letter. We invite you to submit a revised manuscript addressing the Reviewer comments.

Thank you for this interesting contribution to Life Science Alliance. We are looking forward to receiving your revised manuscript.

Sincerely,

B. MANUSCRIPT ORGANIZATION AND FORMATTING:

Reviewer #1 (Comments to the Authors (Required)):

The author's group has previously generated the neuronal-specific conditional p32 knockout (p32cKO) mice and showed decreased mitochondrial function in neurons and oligodendrocytes, which resulted in axonal degeneration and myelin destruction in select brain regions. As the follow-up, the current study by Toshima et al. aimed to explore the molecular mechanism underlying using the same mouse model and additionally validating in three different cell lines. The author found a significant reduction in the expression of several cholesterol synthase genes including Srebp2, a master transcription factor for cholesterol synthesis system. Using pharmacological tools and siRNA, the author suggested that mitochondrial translation inhibition by p32cKO or CAP treatment could activate Pyk2 and Gsk3 β and then regulate protein levels of nuclear Srebp2 (nSrebp2), thus eventually led to downregulation of cholesterol synthase gene expression and caused impaired myelination and synaptogenesis. The overall topic is interesting, however there are major limitations in the current study. Most conclusions are overstated, and the results are insufficient to support the authors' conclusion. Critical positive controls were also missing in a couple of experiments. The overall writing may also be improved.

Main concerns:

1. Only one mouse per group (4 regions in WT and 32 regions in KO mice) was evaluated in Fig 1C. Increased sample size and equal analysis regions are needed to get reliable and unbiased results.
2. All the data from Fig 1A-D are from cerebellar region, while the data from Fig 1E-1F and later figures are from medulla oblongata. Results from the same brain region should be used throughout the manuscript.
3. Page 8 line 194-197, given many cholesterol synthase genes were significantly downregulated, without a specific Srebp2 rescue experiment, it's hard to tell if reduced nSrebp2 expression is directly responsible for the reduction of cholesterol synthesis and "leading to a reduction in synaptic vesicles and demyelination"
4. Undifferentiated SH-SY5Y cells, 3T3-L1 cells, and MEFs may not be the suitable cell models to use in the currently study as neurons and oligodendrocytes showed most phenotypes in the p35cKO mice.
5. In all cell culture experiments with pharmacological manipulation, positive controls are needed to show if the treatment and dose used is successful in the current cell line. For example, mitochondrial translation should be evaluated in cells treated with CAP; cell GSK3 β activity should be examined after treated with GSK3 β inhibitors.
6. Fig 2A and 3D, the mRNA level of Srebp2 should also be included.
7. Overexpression of Srebp2 in CAP treated cells and measure the mRNA levels of other cholesterol synthase gene will provide key evidence if Srebp2 is the upstream regulator of the changes in Fig 2A.
8. Fig 2E, Srebp2 mRNA expression was not downregulated upon mitochondrial translation inhibition in CAP treated cells, which was inconsistent with what was seen in the p35cKO mouse brain. Therefore, the cell model may not recapitulate what is happening in the mouse brain and may not be used as a proper model to study the molecular mechanism in p35cKO mice.
9. nSrebp2, p-GSK3 β , and GSK3 β levels should be shown in both Fig 3B and 3E.
10. nSrebp2, p-GSK3 β , GSK3 β , p-PYK2, Pyk2 levels should be shown in both Fig 4B and 4C.
11. p-PYK2 and Pyk2 levels need to be examined upon treatment of GSK3 β to determine if it's truly in the upstream of the proposed pathway.

Other concerns:

1. The references cited in the current manuscript are mostly published from many years ago. The author should cite more recent paper to reflect current progress and knowledge.
2. In the Introduction, should give a brief background on the function of p32 before introducing p32nKO mice.
3. The current data has little connection to the aging and pathogenesis of neurodegenerative disorders. The author perhaps want to emphasize less on this in the abstract and introduction.
4. Discussion is too long and not pertained to the result of the current data.
5. There are few random * and ns in the bottom right corner in figure 1 and should be removed.

Reviewer #2 (Comments to the Authors (Required)):

The paper focuses on neuron-specific mitochondrial translation impairment that leads to demyelination. Inhibiting mitochondrial translation reduces cholesterol synthase gene expression and degrades Srebp2. The authors found that Pyk2 and Gsk3 impact

the process, and targeting this pathway can restore Srebp2 cholesterol metabolism. The paper is well written, timely and could be of interest when made stronger regarding replicability and addressing some unclarities.

- Line 148. The authors state: "How mitochondrial dysfunction affects Srebp2 expression patterns and gene expression related to cholesterol synthesis remains unclear." Two recent papers link mitochondrial dysfunction and mtDNA leakage/degradation problems paralleled with lysosomal defects to the cGas-STING pathway; PMID: 37532932 (ageing) and PMID: 37225734 (late-onset Alzheimer's disease). Interestingly the AD paper detected an aberrant cholesterol accumulation in lysosomes and srpb2 involvement, with an accumulation of multilamellar bodies. Did the authors detect any similar lysosomal or autophagic/mitophagic pathology on the EM images taken for Figure 1? This, knowing white matter changes have been described in several lysosomal storage disorders as well, making a lysosomal involvement likely.
- Figure 1. Could the authors provide an overview image of panel A and B from which these "insets" were taken? Similarly, could for panel D a square be drawn on the overview picture from which the insets were taken? Like the authors have done for figure S1. It is more representative to have both the overview and zoom.
- Figure 1. The electron microscopy images (panel B-D) and synaptophysin staining (panel A) were performed on n=1 mouse per group. To exclude inter-individual variation, could the authors extend the analysis to n=3. Or perform another complementary analysis that can be scaled up more easily, like a synaptic stain (on primary cultures) of e.g. vGlut1 and GluR1?
- Line 211 and 212. Could the Supplementary figure 2, 3, 4 and 5 be merged as 1, as panels A-D, to allow for a better comparison?
- Line 215. "We investigated whether the decreased expression of the sterol transcription factor nSrebp2 is related to the decreased expression of the cholesterol synthase genes." Given the authors see the protein levels of nSREBP2 in SH-SY5Y cells to already decrease after 24 h of treatment with CAP (Fig. 2 panel B), while on the transcript level there is not yet a significant effect after 24 h (panel A), how can it be concluded that this is a downstream effect?
- Figure 2B. It is not mentioned in the figure legend, but how many times was this repeated? N = ?
- Line 221. "Immunohistochemical staining was also performed to observe changes in the localization of Srebp2 upon CAP treatment." Could the authors also provide an inset/zoom of 1 cell as well as calculate whether the ratio of cytosolic:nuclear Srebp2 might be affected? There might also be an additional localization effect on top of the protein levels? The calculation should be performed on independent experiments. In addition, it would be informative to do a co-localization with Scap and/or an ER marker.
- Line 241. The authors restore expression of cholesterol synthase genes by removing a polyubiquitination enzyme. Should in this case not an effect be expected only on the protein level? How do the authors explain the effect on the transcript levels?
- Line 245. Fig S8. Could this blot be repeated and fold changes quantified to look for replicability?
- Line 246. "These results suggest that, upon the inhibition of mitochondrial translation, nSrebp2 is ubiquitinated by Fbw7 and degraded by the proteasome, resulting in decreased expression of cholesterol synthase genes." The effect of the Fbw7 KD should be substantiated with a more direct and independent method to really conclude this, certainly given the strong effect on the transcriptional level, which does not seem to be in line with an protein Ub-degradation effect. Could the phosphorylation status of SREBP2 be checked for example?
- Line 253: "Studies have also reported that the activation of Gsk3 β phosphorylates a specific site on Srebp1 and suppresses its function (Bengoechea-Alonso & Ericsson, 2009; Dong et al., 2016)". Could the authors extend the blot analysis of Figure 3E to SREBP1? Could some of the observed effects be attributed to a combination of SREBP1 and SREBP2?
- Line 373: "These results suggest that the activation of Pyk2-Gsk3 β through the inhibition of mitochondrial translation is responsible for the increase in phosphorylated tau." I think it would be more correct to say that it "could contribute" to the tau pathology, given the authors did not do rescue experiments on the mice.

Figure legends general remark.

Many of the legends do not contain information on the n-value. If 1 blot or image is shown, is this a representative blot/image of many? Or has this been observed just that once?

Referee Cross-Comments

I agree with both reviewers on the raised concerns; things that need to be addressed.

Reviewer #3 (Comments to the Authors (Required)):

This paper provides experimental data supporting the idea that severe suppression of mitochondrial translation leads to reduced level of cholesterol, and associated synaptic vesicles. This reduction is proposed to be caused by reduction of nuclear (n)Srebp2, a transcription factor involved in the activation of several genes encoding enzymes involved in cholesterol synthesis. The decrease of nSrebp2 is proposed to be caused by increased activating phosphorylation of GSK3, a serine-threonine kinase originally described as a regulator of glycogen metabolism. In turn, Gsk3 activating phosphorylation is proposed to be caused by increased activation through phosphorylation of PIK2, a Ca²⁺-dependent tyrosine kinase. As a final result, the phosphorylation of Srebp2 by PIK2-activated GSK3 ultimately leads to inactivation of Srebp2, its polyubiquitination and proteosomal degradation, resulting in reduced expression of key enzymes for cholesterol system. In the abstract the sentence at line 43 says:"the activation of the PYK2-GSK3 β axis suppresses the ubiquitination of Srebp2 and cholesterol gene expression", but the ubiquitination suppression of Srebp2 should lead to the increase of active Srebp2 and therefore to activation of cholesterol synthesis.

Anyway, there are several examples in which defective mtDNA translation are associated with hypomyelination or dysmyelination leading to leukodystrophy, although I am not sure that cholesterol was specifically studied in these clinical cases. In particular, numerous recessive defects in genes encoding mitochondrial aminoacyl-tRNA synthases are often associated with specific leukodystrophic patterns. These conditions are frequently associated with a clear defect in mitochondrial translation associated with hypomorphic mutations in *ARS2* genes whereas no null mutations have been found in homozygosity, which suggest that the complete absence of an *ARS2* gene expression is embryonic lethal. The association between a progressive, age-related, relative decline of mtDNA translation in old brains and demyelination found in the CNS of adult-onset neurodegenerative disorders, mainly AD, is still rather unclear, and whether this effect, if present, can be ascribed to key metabolic pathways (e.g. cholesterol biosynthesis) or to loss of oligodendrocytes as part of the overall cell degeneration process in the brain is a matter of debate.

The very complex mechanism proposed in the paper is based on the investigation of two main models, a brain specific mouse KO for p32, and cloramphenicol (CAP) treated human cells. The proteins involved in the proposed mechanism all have pleiotropic effects and regulative mechanisms, including p32, a mitochondrial matrix chaperone associated with the physiological maintenance of OXPHOS through number of mechanisms, some of which are involved in mtDNA translation. Pleiotropic effects are retained also by Pyk2 and Gsk3beta, and possibly by a number of other proteins mentioned throughout the text, particularly in the discussion. Therefore a variety of mechanisms can perhaps be considered to interact with these factors, in addition to the Pyk2-Gsk3beta-Srebp2 axis induced by brain ablation of p32 or by CAP exposure of human cells. Both p32 and CAP have drastic effects on mtDNA translation, leading to approximately 50% reduction in the amount of Srebp2 protein in p32 KO brains and a progressively higher decrease with time in CAP-exposed cells. Is this reduction sufficient to explain the partial suppression of cholesterol synthesis? What is the mechanism by which the decline of mtDNA translation determines the activation of Pyk2? These questions remain unanswered.

Nevertheless, the data in this paper convincingly show that in both p32KO brains and in CAP-treated human cells the phosphorylation of both GSK3beta and Pyk2 is significantly increased (fig.3A and 4A) supporting the idea that the decrease of Srebp2 can be associated to these conditions, possibly related to reduced mtDNA translation, by an unknown mechanism. The Authors are unclear when they talk about expression of gene, including SREBP2, because in several panels they show the effects, usually a decrease, in mRNA expression, of several genes, including Srebp2 (for instance in figure 1 E, whereas the overall message the paper conveys is that repression of Srebp2 occurs at the protein level, because of increased degradation of the protein. The Authors should better clarify this point.

In the Discussion and elsewhere, the Authors engage the readership in a complex dissertation about other issues, including the phosphorylation of Tau carried out by Gsk3, the possible increase in MAM-related contacts between mitochondria and the ER, a hypothesized role of Scap, a controller of Srebp2 expression, and other observations that could be released in a more succinct way. In summary I found this paper an interesting contribution, which needs some clarifications in the data display and interpretation. The key point, i.e. the mechanistic link between reduced mtDNA translation and reduced activation of Cholesterol synthesis through Srebp2 activation is still not answered, and the discussion should not present data that are not displayed in the Results section. A quantitative demonstration that the partial reduction observed of Srebp2 amount is effective in reducing cholesterol level and cause hypomyelination is missing. The link between reduced mitochondrial translation and activation of the PK2-Gsk3-Srebp2 is an interesting observation but the mechanism determining this connection is missing as well. Other considerations in the Discussion are not really supported by results displayed in the appropriate section and should be reduced, or associated with experimental evidence.

Chief Editor
Life Science Alliance
25th April 2024

Dear Editor,
“Mitochondrial translation failure represses cholesterol gene expression via Pyk2-Gsk3 β -Srebp2 axis ” LSA-2023-02423

We sincerely appreciate that you provided several critical comments. We are sure that your suggestions were very important for improvement of our manuscript. We added several experiments and extensively improve this revised manuscript according to your comments.

Reviewer #1 (Comments to the Authors (Required)):

The author's group has previously generated the neuronal-specific conditional p32 knockout (p32cKO) mice and showed decreased mitochondrial function in neurons and oligodendrocytes, which resulted in axonal degeneration and myelin destruction in select brain regions. As the follow-up, the current study by Toshima et al. aimed to explore the molecular mechanism underlying using the same mouse model and additionally validating in three different cell lines. The author found a significant reduction in the expression of several cholesterol synthase genes including Srebp2, a master transcription factor for cholesterol synthesis system. Using pharmacological tools and siRNA, the author suggested that mitochondrial translation inhibition by p32cKO or CAP treatment could activate Pyk2 and Gsk3 β and then regulate protein levels of nuclear Srebp2 (nSrebp2), thus eventually led to downregulation of cholesterol synthase gene expression and caused impaired myelination and synaptogenesis.

The overall topic is interesting, however there are major limitations in the current study. Most conclusions are overstated, and the results are insufficient to support the authors' conclusion. Critical positive controls were also missing in a couple of experiments. The overall writing may also be improved.

We sincerely appreciate that you provided several critical comments. We are sure that your suggestions were very important for improvement of our manuscript. We added several experiments and extensively improve this revised manuscript according to your comments.

Main concerns:

1. Only one mouse per group (4 regions in WT and 32 regions in KO mice) was evaluated in Fig 1C. Increased sample size and equal analysis regions are needed to get reliable and unbiased results.

Thank you for reviewing our manuscript and for providing helpful comments.

Electron micrographs cannot be taken in our laboratory and have to be outsourced, which is very costly. Therefore, electron microscopy was only tried on one animal at a time.

In this revision, we increased the number of synaptic observation areas in WT, measuring 20 synapses in WT mice and 40 synapses in p32cKO mice. Then Fig.1C was moved to Fig.1B and modified.

2. All the data from Fig 1A-D are from cerebellar region, while the data from Fig 1E-1F and later figures are from medulla oblongata. Results from the same brain region should be used throughout the manuscript.

We apologize for any confusion this may have caused. Brain slices were prepared from coronal sections of the mouse cerebellum.

Immunostaining, electron microscopy, mRNA and protein extraction were all performed in the white matter of the pons and medulla oblongata immediately below the cerebellum.

In this revised manuscript, the following text should be changed.

In Result section (Page 7)

“To study the state of synapses in the brain, transmission electron microscopy of the white matter of 5-week-old mice in the pons and medulla oblongata revealed a decrease in the number of synaptic vesicles present in neurons of p32cKO mice

We also examined by immunohistochemistry using antibodies against synaptophysin, a synaptic vesicle membrane protein in the white matter region of the pons and medulla oblongata of control and p32cKO mice at 6 weeks of age.”

“white matter region of pons and medulla oblongata”

We also change the Figure legends and supplementary Figure legends in this revised manuscript.

3. Page 8 line 194-197, given many cholesterol synthase genes were significantly downregulated, without a specific Srebp2 rescue experiment, it's hard to tell if reduced nSrebp2 expression is directly responsible for the reduction of cholesterol synthesis and "leading to a reduction in synaptic vesicles and demyelination"

According to the reviewer's comment, we overexpressed 1-468 aa SREBP2 cDNA containing the deduced first methionine site in SH-SY5Y cells for rescue experiments (Fig.3B). Then, the chloramphenicol-induced reduction in cholesterol gene expression was suppressed.

4. Undifferentiated SH-SY5Y cells, 3T3-L1 cells, and MEFs may not be the suitable cell models to use in the currently study as neurons and oligodendrocytes showed most phenotypes in the p32cKO mice.

According to the reviewer's comment, we repeated several experiences including the fusing the differentiated SH-SY5Y cells and Oligodendrocyte cells on this revision.

We used the differentiated SH-SY5Y cells in Fig 2B, Fig 2D, Fig 4C, Fig 5C.

We also used Oligodendrocyte cells in Supplementary Fig S2E, S2H, S2J, S2L, S4A, S5A.

We obtained the same results with undifferentiated and differentiated SH-SY5Y cells and Oligodendrocyte cells.

5. In all cell culture experiments with pharmacological manipulation, positive controls are needed to show if the treatment and dose used is successful in the current cell line. For example, mitochondrial translation should be evaluated in cells treated with CAP; cell GSK3 β activity should be examined after treated with GSK3 β inhibitors.

According to the reviewer's comment, we examined the mitochondrial translation by CAP treatment. We observed that COXI protein levels, which were encoded mtDNA and translated in mitochondria, were significantly reduced by 100 μ M CAP treatment. (Figure 2C, 2D and S2H). Then we used the 100 μ M CAP treatment in this revised manuscript.

In this revised manuscript, we added the COXI result in this revision. (Figure 2C, 2D and S2H).

We also examine whether GSK3 β inhibitor were sufficiency or not, we performed the experience

Then we observed that: The effectiveness of GSK3 inhibitors was confirmed by phosphorylation of Gsk3 β (Ser9) (Fig S4B). (Page 11)

In this revision, we added the sentence in Page 11.

6. Fig 2A and 3D, the mRNA level of Srebp2 should also be included.

According to the reviewer's comment, we examined the experience, then we observed that mRNA level of Srebp2 were reduced in 25%

In this revision, we changed the sentence in Page 8.

” In SH-SY5Y cells treated with CAP, the expression of genes encoding a group of enzymes involved in cholesterol synthesis was reduced to about 40%–50% upon 48 and 72 h of treatment compared with the level in untreated cells and 25% reduction in gene expression of the transcription factor Srebp2 was observed (Fig 2A).”

7. Overexpression of Srebp2 in CAP treated cells and measure the mRNA levels of other cholesterol synthase gene will provide key evidence if Srebp2 is the upstream regulator of the changes in Fig 2A.

According to the reviewer's comment, we performed the overexpression experience in Fig.3B.

In this revision, we added the sentence below (Page 9~10).

“Next, if down-regulation of Srebp2 by CAP suppresses cholesterol gene expression, we examined whether overexpression of nSrebp2 in CAP-treated cells would restore cholesterol synthase gene expression. First, a nuclear transfer type nSrebp2 expression vector was generated and transfected into SH-SY5Y cells, and cholesterol gene expression was examined with and without CAP treatment. Overexpression of nSrebp2 improved cholesterol gene expression, which was suppressed by CAP, indicating that Srebp2 is a master regulator of cholesterol gene expression (Fig S3B and Fig 3B)

8. Fig 2E, Srebp2 mRNA expression was not downregulated upon mitochondrial translation inhibition in CAP treated cells, which was inconsistent with what was seen

in the p32cKO mouse brain. Therefore, the cell model may not recapitulate what is happening in the mouse brain and may not be used as a proper model to study the molecular mechanism in p32cKO mice.

Additional experiments were performed in accordance with the reviewers' comments (Fig. 3C). Of the previously observed changes in Srebp2 mRNA expression due to CAP treatment, no significant differences were observed in the N=1 experiment due to high variability, while a decreasing trend was observed in the other samples. Therefore, when additional experiments were performed, a significant decrease in Srebp2 gene expression due to CAP treatment was observed, similar to the results obtained in the brains of p32cKO mice.

Similarly, a decrease in Srebp2 gene expression due to CAP treatment was observed in the experimental samples in Fig. 4E. This was similar to the result observed in the brains of p32cKO mice. Similar results were also observed in differentiated SH-SY5Y cells and oligodendrocyte cells used in additional experiments (Fig. 2B & Fig. S2E).

9. nSrebp2, p-GSK3 β , and GSK3 β levels should be shown in both Fig 3B and 3E.

According to the reviewer's comment, we performed the several experience. Then we added the results in Fig 4B, 4C 4F in this revision. Then Fig.3B and 3E was moved to Fig.4B,4C and 4F and modified.

10. nSrebp2, p-GSK3 β , GSK3 β , p-PYK2, Pyk2 levels should be shown in both Fig 4B and 4C.

According to the reviewer's comment, we performed the several experience. Then Fig.4B and 4C was moved to Fig.5B,5C and 5D and added the results in this revision.

11. p-PYK2 and Pyk2 levels need to be examined upon treatment of GSK3 β to determine if it's truly in the upstream of the proposed pathway.

According to the reviewer's comment, we performed the several experience. We observed that GSK3 β inhibitor did not affect the Pyk2 phosphorylation.

In this revision. We added the sentence in Page 11~12.

“We next investigated whether activates Gsk3 β using PYK2 inhibitors. Phosphorylated Gsk3 β (Tyr216) expression, increased by CAP, was suppressed by Pyk2 inhibitors (Fig 5D). However, GSK3 β inhibitor did not inhibit the phosphorylation of Pyk2, suggesting that Pyk2 is a kinase upstream of Gsk3 β (Fig S5A). In SH-SY5Y, Pyk2 inhibition was also observed to restore nSrebp2 expression by CAP treatment (Fig 5E). “

Other concerns:

1. The references cited in the current manuscript are mostly published from many years ago. The author should cite more recent paper to reflect current progress and knowledge.

According to the reviewer’s comment, we changed the reference in this revision.

2. In the Introduction, should give a brief background on the function of p32 before introducing p32nKO mice.

According to the reviewer’s comment, we added the p32 function in this revision.

3. The current data has little connection to the aging and pathogenesis of neurodegenerative disorders. The author perhaps want to emphasize less on this in the abstract and introduction.

According to the reviewer’s comment, we changed the abstract and Introduction in this revision.

4. Discussion is too long and not pertained to the result of the current data.

According to the reviewer’s comment, we shortened the Discussion section in this revision.

5. There are few random * and ns in the bottom right corner in figure 1 and should be removed.

According to the reviewer’s comment, we deleted the * and ns in this revision.

Reviewer #2 (Comments to the Authors (Required)):

The paper focuses on neuron-specific mitochondrial translation impairment that leads to demyelination. Inhibiting mitochondrial translation reduces cholesterol synthase gene expression and degrades Srebp2. The authors found that Pyk2 and Gsk3 impact the process, and targeting this pathway can restore Srebp2 cholesterol metabolism. The paper is well written, timely and could be of interest when made stronger regarding replicability and addressing some unclarities.

1, - Line 148. The authors state: "How mitochondrial dysfunction affects Srebp2 expression patterns and gene expression related to cholesterol synthesis remains unclear." Two recent papers link mitochondrial dysfunction and mtDNA leakage/degradation problems paralleled with lysosomal defects to the cGas-STING pathway; PMID: 37532932 (ageing) and PMID: 37225734 (late-onset Alzheimer's disease). Interestingly the AD paper detected an aberrant cholesterol accumulation in lysosomes and srbp2 involvement, with an accumulation of multilamellar bodies. Did the authors detect any similar lysosomal or autophagic/mitophagic pathology on the EM images taken for Figure 1? This, knowing white matter changes have been described in several lysosomal storage disorders as well, making a lysosomal involvement likely.

According to the reviewer's comment, we cited the two papers in this revision paper. We also established cardiomyocyte specific p32knockout mice and published three papers.

PMID: 33528041 PMID: 37793777 PMID: 28498888

We found that the mitochondrial translation-deficient hearts from p32-knockout mice were found to exhibit enlarged lysosomes containing lipofuscin, suggesting impaired lysosome and autolysosome function. We also found that accumulation of multilamellar bodies in lysosome in p32cKO heart.

However, no major lysosomal dysfunction was observed in neuronal-specific p32 knockout mice, probably because they died at 8 weeks of age.

In the future, we would like to generate neuron-specific or oligodendrocyte-specific p32 knockout mice to investigate lysosomal dysfunction.

2. - Figure 1. Could the authors provide an overview image of panel A and B from which these "insets" were taken? Similarly, could for panel D a square be drawn on the

overview picture from which the insets were taken? Like the authors have done for figure S1. It is more representative to have both the overview and zoom.

According to the reviewer's comment, we prepared the overview image and inset image in this revised manuscript. Then Fig.1A, 1C, and 1D was moved to Fig.S1A,1B and 1C and modified.

3. - Figure 1. The electron microscopy images (panel B-D) and synaptophysin staining (panel A) were performed on n=1 mouse per group. To exclude inter-individual variation, could the authors extend the analysis to n=3.

Or perform another complementary analysis that can be scaled up more easily, like a synaptic stain (on primary cultures) of e.g. vGlut1 and GluR1?

According to the reviewer's comment,

Electron micrographs cannot be taken in our laboratory and have to be outsourced, which is very costly. Therefore, electron microscopy was only tried on one animal at a time.

In this revision, we increased the number of synaptic observation areas in WT, measuring 20 synapses in WT mice and 40 synapses in p32cKO mice. Then Fig. 1B was changed.

Immunohistochemical analyses with synaptophysin were performed on six mice (WT, KO, three each) at different numbers of weeks of age (5, 6 and 8 weeks). Examples of 8-week-old mice are shown in the paper. In this revised version, immunostaining has been moved to Supplementary Data S1A.

3.- Line 211 and 212. Could the Supplementary figure 2, 3, 4 and 5 be merged as 1, as panels A-D, to allow for a better comparison?

Thank you for reviewer's comment. In accordance with the reviewers' comments, the supplementary figures were rearranged and changed for consistency.

4. - Line 215. "We investigated whether the decreased expression of the sterol transcription factor nSrebp2 is related to the decreased expression of the cholesterol synthase genes." Given the authors see the protein levels of nSREBP2 in SH-SY5Y

cells to already decrease after 24 h of treatment with CAP (Fig. 2 panel B) , while on the transcript level there is not yet a significant effect after 24 h (panel A), how can it be concluded that this is a downstream effect?

In accordance with the reviewers' comments, we observed that the protein level of nSrebp2 in SH-SY5Y cells was already reduced 24 h after CAP treatment (Fig. 2C), whereas at the transcriptional level, the mRNA levels of cholesterol synthesis genes still showed no significant effect 24 h later (Fig. 2A).

The amount of cholesterol synthesis gene mRNA is controlled by regulation and stability of the transcription level. Therefore, as there is often a time lag in the relationship between transcription factor protein levels and mRNA, the data at 24 h show a decrease in the expression of Srebp2 protein but not of cholesterol synthesis gene mRNA. However, after 48 h, mRNA showed a sharp decrease, indicating that the reduction of mRNA by Srebp2 siRNA and the overexpression of Srebp2 saved the effect of CAP, indicating that the reduction of Srebp2 protein affected mRNA.

5.- Figure 2B. It is not mentioned in the figure legend, but how many times was this repeated? N = ?

In accordance with the reviewers' comments, we put the number of experiments into all Figure legends in this revision manuscript.

6. - Line 221. "Immunohistochemical staining was also performed to observe changes in the localization of Srebp2 upon CAP treatment." Could the authors also provide an inset/zoom of 1 cell as well as calculate whether the ratio of cytosolic :nuclear Srebp2 might be affected? There might also be an additional localization effect on top of the protein levels? The calculation should be performed on independent experiments. In addition, it would be informative to do a co-localization with Scap and/or an ER marker.

Immunohistochemical staining was also performed to observe changes in Srebp2 localization due to CAP treatment. In this revision, similar experiments were performed on SH-SY5Y cells and oligodendrocyte cells, and high (Fig. 2E & S2L) and low (Fig. S2I & S2J) magnification views of the cells are also shown. Co-localization with the ER marker calnexin confirmed that the majority of Srebp2 was localized to the nucleus (Fig. 2E & S2L).

It was also confirmed that CAP treatment reduced Srebp2 in the nucleus. Nuclear

fluorescence intensity was calculated for 10 cells in three independent experiments.

7. - Line 241. The authors restore expression of cholesterol synthase genes by removing a polyubiquitination enzyme. Should in this case not an effect be expected only on the protein level? How do the authors explain the effect on the transcript levels?

Sorry for the lack of explanation.

In normal cells, the transcription factor Srebp2 is phosphorylated by various stimuli and degraded by ubiquitin proteasomes. The expression of cholesterol synthesis genes is promoted by the protein level of nSrebp2. Therefore, when the protein level of Srebp2 is rescued by Fbw7 siRNA, cholesterol synthase gene expression should be restored and overexpression of Srebp2 restores cholesterol synthase gene expression. These results suggest that the protein level of nSrebp2 affects gene expression.

8.- Line 245. Fig S8. Could this blot be repeated and fold changes quantified to look for replicability?

In accordance with the reviewers' comments, we have changed Fig. S8 in this revised manuscript. Also, Fig.S8 was moved to Fig.S3C and modified.

We have presented Fig.S3C we performed the triplicated experience and shoe the figure legend in this revised manuscript.

9. - Line 246. "These results suggest that, upon the inhibition of mitochondrial translation, nSrebp2 is ubiquitinated by Fbw7 and degraded by the proteasome, resulting in decreased expression of cholesterol synthase genes." The effect of the Fbw7 KD should be substantiated with a more direct and independent method to really conclude this, certainly given the strong effect on the transcriptional level, which does not seem to be in line with an protein Ub-degradation effect. Could the phosphorylation status of SREBP2 be checked for example?

According to reviewer's comment, we changed the text, because this siRNA experiment alone does not directly detect Fbw7.

In page 10

"This suggests that inhibition of mitochondrial translation reduces Srebp2 protein levels and involves the ubiquitinating enzyme Fbw7."

We tried to investigate the phosphorylation of Srebp2, but this was not possible this time due to the lack of phosphorylation antibodies and the lack of good antibodies that can do IP, but in the future we would like to identify the phosphorylation sites as well.

10. - Line 253: "Studies have also reported that the activation of Gsk3 β phosphorylates a specific site on Srebp1 and suppresses its function (Bengoechea-Alonso & Ericsson, 2009; Dong et al., 2016)". Could the authors extend the blot analysis of Figure 3E to SREBP1? Could some of the observed effects be attributed to a combination of SREBP1 and SREBP2?

We investigated the expression of nSrebp1 after CAP treatment using Srebp1 antibodies. It was observed that nSrebp1 levels decreased with CAP administration and were found to be restored with GSK inhibitors. The results were the same as for nSrebp2. Therefore, Fig S4C was added. Error bars represent mean \pm SD of three independent experiments.

11.- Line 373: "These results suggest that the activation of Pyk2-Gsk3 β through the inhibition of mitochondrial translation is responsible for the increase in phosphorylated tau." I think it would be more correct to say that it "could contribute" to the tau pathology, given the authors did not do rescue experiments on the mice.

Due to reviewers' comments, the above text has been deleted in this revised version. This is due to the paucity of data on tau.

12. Figure legends general remark.

Many of the legends do not contain information on the n-value. If 1 blot or image is shown, is this a representative blot/image of many? Or has this been observed just that once?

Due to reviewers' comments, we added the number of experience in this revised manuscript.

Referee Cross-Comments

I agree with both reviewers on the raised concerns; things that need to be addressed.

Reviewer #3 (Comments to the Authors (Required)):

This paper provides experimental data supporting the idea that severe suppression of mitochondrial translation leads to reduced level of cholesterol, and associated synaptic vesicles. This reduction is proposed to be caused by reduction of nuclear (n)Srebp2, a transcription factor involved in the activation of several genes encoding enzymes involved in cholesterol synthesis. The decrease of nSrebp2 is proposed to be caused by increased activating phosphorylation of GSK3, a serine-threonine kinase originally described as a regulator of glycogen metabolism. In turn, Gsk3 activating phosphorylation is proposed to be caused by increased activation through phosphorylation of PYK2, a Ca²⁺-dependent tyrosine kinase. As a final result, the phosphorylation of Srebp2 by PYK2-activated GSK3 ultimately leads to inactivation of Srebp2, its polyubiquitination and proteosomal degradation, resulting in reduced expression of key enzymes for cholesterol system.

1. In the abstract the sentence at line 43 says: "the activation of the Pyk2-Gsk3 β axis suppresses the ubiquitination of Srebp2 and cholesterol gene expression", but the ubiquitination suppression of Srebp2 should lead to the increase of active Srebp2 and therefore to activation of cholesterol synthesis.

Sorry for the confusion.

As a result, mitochondrial translation defects activate Pyk2- Gsk3 β phosphorylation and active Gsk3 β decreases nSrebp2 protein by the ubiquitin-proteasome pathway. nSrebp2 decrease is associated with suppressed cholesterol gene expression. Therefore, the abstract was changed in this revision.

.” Activation of the Pyk2-Gsk3 β axis is involved in the ubiquitination of nSrebp2 and reduces nSrebp2 protein, resulting in suppression of cholesterol gene expression.”

2. Anyway, there are several examples in which defective mtDNA translation are associated with hypomyelination or dysmyelination leading to leukodystrophy, although I am not sure that cholesterol was specifically studied in these clinical cases. In particular, numerous recessive defects in genes encoding mitochondrial aminoacyl-tRNA synthases are often associated with specific leukodystrophic patterns. These conditions are frequently associated with a clear defect in mitochondrial translation associated with hypomorphic mutations in ARS2 genes whereas no null mutations have been found in homozygosity, which suggest that the complete absence

of an ARS2 gene expression is embryonic lethal.

It is agreed that mitochondrial translation abnormalities, such as ARS2 gene and p32 dysfunction, are associated with the white matter dystrophy phenotype and cholesterol was specifically studied in these cases. However, it was reported that cholesterol influences myelination at many steps, from the differentiation of myelinating glial cells, over the process of myelin membrane biogenesis, to the functionality of mature myelin (Gesine Saher 2015). The experimental results suggest that mitochondrial translation abnormalities are associated with reduced expression of cholesterol genes, which in part leads to demyelination and synaptic dysfunction. One of the causes of leukoencephalopathy is thought to be reduced cholesterol gene expression and associated demyelinating changes due to mitochondrial translation abnormalities. In this revised manuscript, we toned down our hypothesis in Figure 6 and changed some sentence in Discussion section.

3. The association between a progressive, age-related, relative decline of mtDNA translation in old brains and demyelination found in the CNS of adult-onset neurodegenerative disorders, mainly AD, is still rather unclear, and whether this effect, if present, can be ascribed to key metabolic pathways (e.g. cholesterol biosynthesis) or to loss of oligodendrocytes as part of the overall cell degeneration process in the brain is a matter of debate.

Thank you for reviewer's comment. We agreed that comment. It is still unclear and whether this effect, if any, is due to major metabolic pathways (e.g. cholesterol biosynthesis), or due to the loss of oligodendrocytes as part of the overall cellular degenerative process in the brain is a matter of debate.

We also suggest that the reduced expression of cholesterol synthesis genes found in the senile brain alone does not correctly explain Alzheimer's disease and may be partly involved. Furthermore, we believe that lysosomal disorders from mitochondrial damage, such as those observed in the cardiac-specific p32cKO, are additively involved in the development of age-related diseases. These are issues that need to be addressed in the future.

We added some sentence in Discussion in this revised manuscript.

4. The very complex mechanism proposed in the paper is based on the investigation of two main models, a brain specific mouse KO for p32, and cloramphenicol (CAP)

treated human cells. The proteins involved in the proposed mechanism all have pleiotropic effects and regulative mechanisms, including p32, a mitochondrial matrix chaperone associated with the physiological maintenance of OXPHOS through number of mechanisms, some of which are involved in mtDNA translation.

Pleiotropic effects are retained also by Pyk2 and Gsk3beta, and possibly by a number of other proteins mentioned throughout the text, particularly in the discussion.

Therefore a variety of mechanisms can perhaps be considered to interact with these factors, in addition to the Pyk2-Gsk3beta-Srebp2 axis induced by brain ablation of p32 or by CAP exposure of human cells. Both p32 and CAP have drastic effects on mtDNA translation, leading to approximately 50% reduction in the amount of Srebp2 protein in p32 KO brains and and a progressively higher decrease with time in CAP-exposed cells. Is this reduction sufficient to explain the partial suppression of cholesterol synthesis? What is the mechanism by which the decline of mtDNA translation determines the activation of Pyk2? These questions remain unanswered.

Thank you for the comment, we understand that the functions of p32 are diverse and cannot be explained solely by the mitochondrial disorder - the reduction of cholesterol synthesis by Pyk2-Gsk3 β -Srebp2. The present paper only partially demonstrates that these systems are involved, but the whole picture will be the subject of future research. Neuronal or oligodendrocyte-specific mice need to be created to confirm which cells are involved in these hypotheses.

Thank you for the comment, we understand that the functions of Pyk2 and Gsk3 β are diverse and cannot be explained solely by the mitochondrial disorder. In near future we have to explain our hypothesis in near future.

According to the comment

Q Is this reduction sufficient to explain the partial suppression of cholesterol synthesis?

Thank you for comment, it is very difficult question, and we did not explain the reduced nSrebp2 sufficient to explain the partial suppression.

In this revised manuscript, we observed the overexpression of nSrebp2 rescue the CAP mediated cholesterol gene suppression, suggesting that reduction of Srebp2 is partially involved suppression of cholesterol gene expression.

Q. What is the mechanism by which the decline of mtDNA translation determines the activation of Pyk2?

Thank you for comment, it is very difficult question, P32(C1qbp) may modulate Ca^{2+} levels in the mitochondrial matrix (Jiang J, Zhang Y,PNAC 1999) PDH, NAD-isocitrate dehydrogenase and oxoglutarate dehydrogenase are all regulated by intramitochondrial Ca^{2+} levels either directly or indirectly. C1QBP may thereby regulate mitochondrial OXPHOS by modulating Ca^{2+} concentrations. These results suggest that loss of p32 disrupts Ca^{2+} homeostasis within the mitochondrial matrix, affecting the mitochondrial exterior and leading to Pyk2 activation. Future studies will investigate whether mitochondrial translation defects affect Pyk2 phosphorylation in detail.

5. Nevertheless, the data in this paper convincingly show that in bot p32KO brains and in CAP-treated human cells the phosphorylation of both GSK3beta and Pyk2 is significantly increased (fig.3A and 4A) supporting the idea that the decrease of Srebp2 can be associated to these conditions, possibly related to reduced mtDNA translation, by an unknown mechanism.

The Authors are unclear when they talk about expression of gene, including SREBP2, because in several panels they show the effects, usually a decrease, in mRNA expression, of several genes, including Srebp2 (for instance in figure 1 E, whereas the overall message the paper conveys is that repression of Srebp2 occurs at the protein level, because of increased degradation of the protein.

It is plausible that a decrease in Srebp2 mRNA leads to a decrease in Srebp2 protein levels. The decrease in Srebp2 protein levels therefore suggests that the decrease in mRNA and the increase in protein degradation are closely linked. However, full-length Srebp2 remains in the Golgi after translation, where it is cleaved by proteases in response to cholesterol levels, translocated to the nucleus and functions as a transcription factor. In other words, nSrebp2 is mainly regulated at the protein level, and in this paper the observed increase in proteolysis led to a decrease in cholesterol gene expression. As a decrease in mRNA cannot be ruled out, it was noted in the paper that a decrease in nSrebp2 protein due to a decrease in Srebp2 mRNA and an increase in nSrebp2 proteolysis was associated with reduced cholesterol gene expression in this paper.

We added the sentence in Page 10.

“SREBP-2 is regulated at the transcriptional and post-translational levels, and specific signaling pathways may be involved in this regulation. We tested whether Srebp2 is

regulated at the post-translational level, as the decrease in Srebp2 protein was greater than the decrease in Srebp2 mRNA.”

We added the sentence in page 13 in Discussion

“The decrease in Srebp2 was associated with decreased expression of Srebp2 mRNA and increased degradation of Srebp2 protein. In this study, the mechanism of Srebp2 protein degradation was analyzed in detail and the Pyk2-Gsk3 β axis was found to be involved.”

6. In the Discussion and elsewhere, the Authors engage the readership in a complex dissertation about other issues, including the phosphorylation of Tau carried out by Gsk3, the possible increase in MAM-related contacts between mitochondria and the ER, a hypothesized role of Scap, a controller of Srebp2 expression, and other observations that could be released in a more succinct way. In summary I found this paper an interesting contribution, which needs some clarifications in the data display and interpretation.

Thank you for the comment,

We changed the Discussion section in revised manuscript a concise wording.

7. The key point, i.e. the mechanistic link between reduced mtDNA translation and reduced activation of Cholesterol synthesis through Srebp2 activation is still not answered, and the discussion should not present data that are not displayed in the Results section. A quantitative demonstration that the partial reduction observed of Srebp2 amount is effective in reducing cholesterol level and cause hypomyelination is missing.

The link between reduced mitochondrial translation and activation of the PYK2-Gsk3-Srebp2 is an interesting observation but the mechanism determining this connection is missing as well. Other considerations in the Discussion are not really supported by results displayed in the appropriate section and should be reduced, or associated with experimental evidence.

Thank you for your comments,

In this revised version, we have chosen to present the expression in a restrained manner due to the weak quantitative evidence that the observed partial reduction in Srebp2 levels is effective in reducing cholesterol levels and causing hypomyelination.

In this revised version, we have then presented several experiences of reduced

mitochondrial translation and activation of Pyk2- Gsk3 β -Srebp2 in SH-SY5Y cells. We have also removed the Discussion section in this revision.

April 25, 2024

RE: Life Science Alliance Manuscript #LSA-2023-02423R

Dr. Takeshi Uchiumi
Kyushu University
Clinical Chemistry and Laboratory Medicine
3-1-1 Maidashi
Fukuoka, Fukuoka 812-8582
Japan

Dear Dr. Uchiumi,

Thank you for submitting your revised manuscript entitled "Mitochondrial translation failure represses cholesterol gene expression via Pyk2-Gsk3 β -Srebp2 axis". We would be happy to publish your paper in Life Science Alliance pending final revisions necessary to meet our formatting guidelines.

- please be sure that the authorship listing and order is correct
- please add the Twitter handle of your host institute/organization as well as your own or/and one of the authors in our system
- please use the [10 author names et al.] format in your references (i.e., limit the author names to the first 10)
- the Methods from the Supplementary file should be incorporated into the main Materials & Methods section

FIGURE CHECKS:

- you may want to consider uploading Figure 6 as a Graphical Abstract rather than as a figure, but this is up to you

A. FINAL FILES:

B. MANUSCRIPT ORGANIZATION AND FORMATTING:

Thank you for your attention to these final processing requirements. Please revise and format the manuscript and upload materials within 4 days.

Sincerely,

April 29, 2024

RE: Life Science Alliance Manuscript #LSA-2023-02423RR

Dr. Takeshi Uchiumi
Kyushu University
Clinical Chemistry and Laboratory Medicine
3-1-1 Maidashi
Fukuoka, Fukuoka 812-8582
Japan

Dear Dr. Uchiumi,

Thank you for submitting your Research Article entitled "Mitochondrial translation failure represses cholesterol gene expression via Pyk2-Gsk3 β -Srebp2 axis". It is a pleasure to let you know that your manuscript is now accepted for publication in Life Science Alliance. Congratulations on this interesting work.

DISTRIBUTION OF MATERIALS:

Again, congratulations on a very nice paper. I hope you found the review process to be constructive and are pleased with how the manuscript was handled editorially. We look forward to future exciting submissions from your lab.

Sincerely,
